# Correcting for water vapor diffusion in air bag samples for isotope composition analysis: case study with drone-collected samples

Di Wang[1,2,3*], Camille Risi[2], Lide Tian[1], Di Yang[1], Gabriel J. Bowen[4], Siteng Fan[2,5],

Yang Su[6] , Hongxi Pang[7], Laurent Z.X Li[2]

[1] Institute of International Rivers and Eco-security, Yunnan University, Yunnan Key Laboratory of International Rivers and Transboundary Eco–security, Kunming 650500, Yunnan, China

[2] Laboratoire de Météorologie Dynamique, IPSL, CNRS, Sorbonne Université, Campus Pierre et Marie Curie, Paris 75005, France

[3] Laboratoire Atmosphères, Observations Spatiales, IPSL, CNRS, UVSQ, Sorbonne Université, Guyancourt 78280, France

[4] Department of Geology and Geophysics, University of Utah, Salt Lake City, Utah 84108, USA

[5] Department of Earth and Space Sciences, Southern University of Science and Technology, Shenzhen 518055, China

[6] Département d'Informatique, École normale supérieure – PSL, 45 Rue d'Ulm, 75005 Paris, France

[7] Key Laboratory of Coast and Island Development of Ministry of Education, School of Geography and Ocean Science, Nanjing University, Nanjing 210023, China

*Corresponding author: di.wang@latmos.ipsl.fr; shidi.upmc@gmail.com

## **Abstract**

Mass spectrometry and laser spectroscopy have been widely employed for precise water vapor isotope measurements. Nevertheless, these techniques are limited by logistical challenges in fieldwork, consequently constraining the temporal and spatial resolution of measurements. Specifically, water vapor isotope measurements are primarily limited to near-surface levels, while measurements in the free troposphere are notably scarce. Portable sampling devices, such as air bags and glass bottles, have therefore become necessary alternatives for collecting, storing, and transporting gaseous samples in diverse environments prior to analysis with less portable instruments. In drone-based high-altitude vapor sampling, air bags are preferred for their lighter weight and greater flexibility compared to glass bottles. Nevertheless, they present specific challenges, such as potential sample contamination and isotopic fractionation during storage, primarily due to the inherent permeability of air bags. Here, we developed a theoretical model for water vapor diffusion through the sampling bag surface, with parameters calibrated through laboratory experiments. This model enables the reconstruction of the initial isotopic composition of sampled vapor based on measurements obtained within the bag and from the surrounding environment. This diffusion model underwent rigorous validation through experiments conducted under varying humidity and isotopic composition differences between the inside and outside of the air bag, confirming its reliability. We applied this correction method to air samples collected at various pressures up to the upper troposphere using an air bag-mounted drone that we developed, thereby estimating the initial isotopic composition and uncertainty based on our observations. Our correction method enhances the reliability and applicability of water vapor isotope observations conducted using drones equipped with air bags, and provides a detailed assessment of all potential sources of error and quantifies the uncertainty range of the observations. This approach leverages the strengths of drone-based air bag sampling while mitigating its limitations, thus facilitating the convenient collection of isotopic data throughout the troposphere.

# 1 Introduction

Water vapor isotopes provide unique insights into the transport, mixing, and phase changes of water in the environment, which are crucial for improving understanding of the climate system, hydrological cycle, atmospheric dynamics, and paleoclimate proxies. Water isotopes have also been applied in climate modeling, weather prediction, and water resource management (Bowen et al., 2019; Galewsky et al., 2016; Gat, 1996; West et al., 2009).

Water isotope analysis has traditionally relied on mass spectrometry, which, while accurate (Ghosh and Brand, 2003; Muccio and Jackson, 2009), demands labor-intensive preparation and lacks portability (West et al., 2010). Methods like cryogenic trapping effectively collect water vapor (Grootes and Stuiver, 1997; Michener and Lajtha, 2008; Steen-Larsen et al., 2011; Yu et al., 2015), but they require long sampling periods, limiting observation scope and timing. Over the past three decades, laser spectroscopy methods such as Cavity Ring-Down Spectroscopy (CRDS)(Hodges and Lisak, 2006) and Off-Axis Integrated Cavity Output Spectroscopy (OA-ICOS) (Johnson et al., 2011) have emerged, delivering a best-in-class combination of speed, high precision, and continuous measurements even in challenging environments such as high altitudes or arid regions with low water vapor content. Advances in these instruments have significantly expanded the field of water isotope research. However, their heavy instrumentation, substantial power requirements, and limited mobility restrict their usability in certain situations, particularly for aerial water vapor isotope measurements, which require lightweight and flexible sampling approaches. As a result, the collection and storage of physical samples are still necessary, increasing the demand for more convenient and efficient sample acquisition methods. Air bags and glass bottles have been practical solutions for collecting, storing, and transporting gaseous samples from various settings (Rozmiarek et al., 2021).

Given that air bags can reduce the weight of sampling equipment and increase sampling flexibility, there is considerable interest in using them for vapor sample collection. This is particularly advantageous for small equipment like drones, where minimizing payload weight is essential for sampling at high altitudes or over long distances. This selection also makes it easier to transport samples and reduces the risk of breakage. However, concerns have arisen regarding the suitability of various sampling materials for storing these samples, primarily due to potential water diffusion through container walls. Diffusion issues are commonly observed in sampling bags during water vapor isotope analysis and have persisted as a longstanding challenge in the field (Gralher et al., 2021; Havranek et al., 2020; Herbstritt et al., 2023; Magh et al., 2022). In previous studies, this issue is particularly concerning in the Direct Vapor Equilibrium Laser Spectroscopy (DVE-LS) method, which has been widely used to rapidly collect and measure water isotopes in evaporation-prone soil, rock, or plant samples. The DVE-LS method simplifies preparation and increases sample throughput by directly analyzing the vapor phase, thus eliminating the need for extensive physical extractions (Gralher et al., 2021; Hendry et al., 2015; Millar et al., 2018; Sprenger et al., 2015; Wassenaar et al., 2008). However, water vapor molecules may exchange between the air inside the bag and the external ambient air during sample storage. Water vapor molecules typically diffuse from areas of high humidity to drier areas. In this process, heavier isotopologues (e.g., $H_2^{18}O$ and HDO) move more slowly than lighter isotopologues (e.g., $H_2^{16}O$) due to their greater mass, resulting in preferential

diffusion of lighter isotopes. This differential diffusion, can alter the original isotopic composition of the collected air samples. Moreover, differential diffusion can also occur due to gradients in isotopic composition.

To mitigate these issues, specific diffusion-tight materials are highly recommended for water vapor isotope measurements, although adsorption effects may still occur with such materials(Herbstritt et al., 2023). Further research and development are still necessary. This may involve exploring alternative materials for more impermeable sampling bags, improving sealing methods to better isolate sampled air, and developing sampling techniques less susceptible to diffusion. Resolving these issues is essential for ensuring the reliability of water vapor isotope measurements using air bags and for accurately understanding atmospheric and hydrological processes.

In light of the ongoing development and further refinement of these techniques and the associated cost constraints, we developed a diffusion model with parameters calibrated through laboratory experiments. This model is capable of assessing the permeability of the air bags and correcting the obtained isotope measurements to the initial pre-diffusion values based on the humidity, isotope values inside and outside the bag, and the sample storage time. This diffusion model was validated through experiments under varying humidity and isotopic composition differences between the inside and outside of the air bag, confirming its reliability. Furthermore, we also applied this diffusion method to air samples collected at different altitudes using a drone-based atmospheric vapor sampling device we developed, to estimate the initial isotope composition and uncertainty. The primary objective of this drone-based field campaign is to obtain atmospheric water vapor isotope data along vertical profiles in the troposphere, providing higher temporal and spatial resolution than satellite observations. The corrected near-surface drone-based measurements using our diffusion model show improved consistency with direct, in-situ surface-level measurements using the Picarro analyzer. Our model, validated by laboratory experiments under a wide range of environmental conditions, was used to assess all potential error sources and to quantify a conservative, comprehensive uncertainty range for the corrected vertical profiles. Although we minimized certain diffusion-related biases, such as pressure in different altitudes and long storage times, temperature and the pressure difference between the inside and outside of the bags on bag permeability were only addressed indirectly via conservative uncertainty estimates. This treatment is justified under our experimental conditions, which are characterized by the short residence time at high altitudes, the reduced diffusivity at low temperatures, and the flexibility of the bags maintaining equal internal and external pressure. Future applications should consider the specific experimental conditions, and future developments may extend the model to explicitly incorporate the dependence of bag permeability on partial pressure and temperature, thereby improving its applicability under a wider range of atmospheric conditions.

## 2   Theoretical basis of diffusion model

### Diffusion model description

Storing vapor samples in air bags prior to isotope measurement may alter the isotopic

2.1

composition of the water vapor. The main reason is the diffusion of water molecules between
the interior and exterior of the air bags, primarily due to the permeability of the bag materials.
We present a mathematical model for the diffusion and fractionation of isotopes across the
surface of the sampling bag. In this model, we assume the ambient vapor flux entering the air
bag changes the internal humidity and vapor isotopic composition, influenced by the different
humidity and water isotopic composition inside and outside the bag (Fig.1).
The flux of water into the bag, F (in $kg/m^2/s$), is expressed as:
$$F = k * (q_e - q(t)) \tag{1}$$
where $q(t)$ represents the variation of humidity inside the air bag over time (in kg/kg), $q_e$
denotes the environmental humidity (in kg/kg), k is water vapor conductance. This first-order
formulation is derived from Fick's diffusion law(Fick, 1855).
Similarly, the flux of isotopologue, $F_i$, either $H_2^{18}O$ or HDO, moving into the bag can be
described as:
$$F_i = k_i * (R_e * q_e - R(t) * q(t)) \tag{2}$$
In this equation, $k_i$ represents the conductance specific to each isotopologue (in $kg/m^2/s$), $R_e$
denotes the isotopic ratio in the environment, and $R(t)$ is the variation of isotopic ratio within
the air bag with time. Notably, the fractionation coefficient here can be denoted as:
$$\alpha = \frac{k}{k_i} \tag{3}$$
Since k represents the total conductance dominated by the lighter molecules ($H_2^{16}O$), and $k_i$
is defined as the conductance of the heavier isotopologues (either $H_2^{18}O$ or HDO), Eq. (3) is
therefore consistent with the conventional definition of fractionation factors ( $\alpha = \frac{k_{light}}{k_{heavy}}$).
The validity of Eqs.1 and 2 relies on the assumptions that internal and external pressures
remain equal to atmospheric pressure, ensuring no pressure gradient across the bag membrane,
that the internal vapor is well-mixed, and that the exchange rate follows a first-order process.
Additionally, if the temperature remains constant, k and $k_i$ are assumed to be constant.

Assuming that diffusion through the bag is the primary transport mechanism, and
neglecting adsorption, the temporal change in humidity can be modeled by the following
differential equation:
$$\frac{d(q(t)*M)}{dt} = F * A \tag{4}$$
where A represents the exchange area (surface area of the air bag), and M is the air mass
inside the bag.
Assuming that M is constant, which is reasonable given that the total mass variation due
to water vapor flux is at most 1%, this equation simplifies to:
$$\frac{dq(t)}{dt} = \frac{F*A}{M} = \frac{k*A}{M} * (q_e - q(t)) \tag{5}$$
Here, we define the water vapor exchange coefficient as follows, it describes the rate at
which water vapor exchanges across the bag membrane:
$$\lambda = \frac{k*A}{M} \tag{6}$$
Similarly, the temporal change in isotopic ratio can be modeled by the following
differential equation:
$$\frac{d(R(t)*q(t)*M)}{dt} = M * (q(t) * \frac{dR(t)}{dt} + R(t) * \frac{dq(t)}{dt})$$

$$= F_i * A = k_i * (R_e * q_e - R(t) * q(t)) * A \tag{7}$$

This equation can be simplified as:
$$\frac{dR(t)}{dt} = \frac{ki*(R_e*q_e-R(t)*q(t))*A/M - R(t)*\frac{dq(t)}{dt}}{q(t)}$$

$$= \frac{\frac{k*A}{M*\alpha}*(R_e*q_e-R(t)*q(t)) - R(t)*\frac{k*A}{M}*(q_e-q(t))}{q(t)}$$

$$= \frac{\frac{\lambda}{\alpha}*(R_e*q_e-R(t)*q(t)) - R(t)*\lambda*(q_e-q(t))}{q(t)}$$

$$= \frac{\lambda}{\alpha} * (R_e - R(t)) + \frac{\lambda}{q(t)} * (q_e - q(t)) * (\frac{R_e}{\alpha} - R(t)) \tag{8}$$


If $q_e$ is stable (the environment is treated as an infinite reservoir), and that k and A are
constant, the differential equation for humidity (Eq. 5) can be analytically solved:
$$q_e - q(t) = (q_e - q_0) * e^{(-\frac{k*A}{M}*t)} = (q_e - q_0) * e^{(-\lambda*t)} \tag{9}$$

where $q_0$ is the initial humidity at t = 0. This equation can also be expressed in terms of
natural logarithms as:
$$\ln(q_e - q(t)) = \ln(q_e - q_0) - \lambda * t \tag{10}$$

Consequently, the slope of $\ln(q_e - q(t))$ against time is $\lambda$.

For the isotopic ratio, the analytical solution is only feasible when the initial humidity
equals the environmental humidity ($q_0 = q_e$):
$$\frac{dR(t)}{dt} = \frac{k*A}{M*\alpha} * (R_e - R(t)) \tag{11}$$

Assuming that $R_e$, $q_e$, k, $\alpha$ and A are constant:
$$\text{Thus, } R_e - R(t) = (R_e - R_0) * e^{(-\frac{k*A}{M*\alpha}*t)} \tag{12}$$

where $R_0$ denotes the initial isotopic ratio at t = 0. Again, taking the natural logarithm, we
obtain:
$$\ln(R_e - R(t)) = \ln(R_e - R_0) - \frac{\lambda}{\alpha} * t \tag{13}$$

This equation demonstrates that the slope of $\ln(R_e - R(t))$ against time is the water vapor
isotopic exchange coefficient $\frac{\lambda}{\alpha}$. Knowing $\lambda$, we can deduce the isotopic fractionation
coefficient $\alpha$ for each isotope.
However, when the environmental humidity differs from the initial humidity inside the air
bag, a numerical solution is required to solve the differential equation for R (Eq. 8).

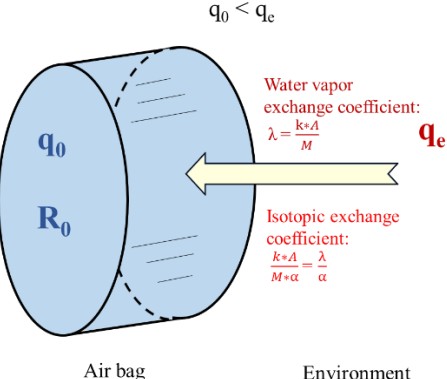

*Figure 1 Schematic illustrating the diffusion model.* $q_0$ *represents the initial humidity in the air bag at t = 0, $q_e$ denotes the environmental humidity, $R_0$ indicates the initial isotopic ratio in the air bag at t = 0, $R_e$ represents the isotopic ratio in the environment, k is water vapor conductance, A denotes the exchange area (surface area of the bag), and M is the mass of air within the bag, and $\alpha$ denotes the isotopic fractionation coefficient.*

### Reconstructing initial water vapor isotopic compositions

The isotopic composition of the air bag water vapor undergoes an exponential evolution over time (Eq. 12). This method of applying exponential evolution equations to reconstruct isotopic compositions has been used in environmental forensics to investigate shifts in water isotopes due to metabolic changes, environmental conditions, or diet (Ayliffe et al., 2004; Cerling et al., 2006). Similarly, in climatology, this method helps retrieving initial isotopic composition from samples such as ice cores or tree rings, affected by evaporation, precipitation, and temperature fluctuations, facilitating historical climate reconstruction (Brienen et al., 2016). Here we apply a similar method, and apply the analytical solution of Eq.8 using data from experiments in which the condition that $q_0$ equals $q_e$ is met to determine the equation parameters.

The constants ($\lambda$, $\alpha^{18}O$, $\alpha^2H$) can be determined through laboratory experiments and Eqs. 10 and 13 (see Subsection 3.2 and 4.1). If we know the initial values within the air bag ($q_0$, $R^{18}O_0$, $R^2H_0$), the ambient values ($q_e$, $R^{18}O_e$, $R^2H_e$) during storage (approximated using ground-level conditions), and the storage time after sampling ($T_{storage}$), we are able to simulate the variations in humidity and isotopic ratios inside the air bag according to Eqs. 5 and 8. Similarly, if we know $T_{storage}$, the humidity and isotopic composition at time t = $T_{storage}$ ($q_{(Tstorage)}$, $R^{18}O_{(Tstorage)}$, $R\_^2H_{(Tstorage)}$ in the air bag, and the ambient values, we can deduce the initial values in the air bag at t = 0 by back-calculating. The equation used for reconstructing the initial isotope ratio ($R_0$) is:

$$R_0 = R_{measured} - \int_0^{T_{storage}} \frac{dR(t)}{dt} dt$$

$$= R_{measured} - \int_0^{T_{storage}} \left( \frac{\lambda}{\alpha} * \left(R_e - R(t)\right) + \frac{\lambda}{q(t)} * \left(q_e - q(t)\right) * \left(\frac{R_e}{\alpha} - R(t)\right) \right) dt \quad (14)$$

where $R_0$ represents the initial isotopic ratio to be reconstructed, $R_{measured}$ is the observed isotopic ratio after $T_{storage}$, and $\frac{dR(t)}{dt}$ is defined in Eq. 8.

This approach allows us to correct for diffusion-induced isotopic shifts and reconstruct the

original vapor composition.

For mathematical clarity and consistency, isotopic ratios (R) are used in the equations presented in previous sections. Replacing R with δ-values would only shift the physical basis without affecting the mathematical validity of the equations or the estimation of α, as the standard ratio cancels out. For clearer visualization, δ-values are used for numerical applications and in the subsequent figures and tables.

The δ and d-excess values used in this study follow standard definitions:

$$\delta^{18}O \ = \left( \frac{R^{18}O_{sample}}{R^{18}O_{standard}} - 1 \right) * 1000 \tag{15}$$

$$\delta^2H \ = \left( \frac{R^2H_{sample}}{R^2H_{standard}} - 1 \right) * 1000 \tag{16}$$

$$d - excess = \ \delta^2H - 8 * \delta^{18}O \tag{17}$$

Here, $R^{18}O_{standard} = 2.0052*10^{-3}$ and $R^2H_{standard} = 1.5576 * 10^{-4}$, corresponding to the VSMOW international reference values.

# 3 Methods and data

## 3.1 Air bag isotope measurements

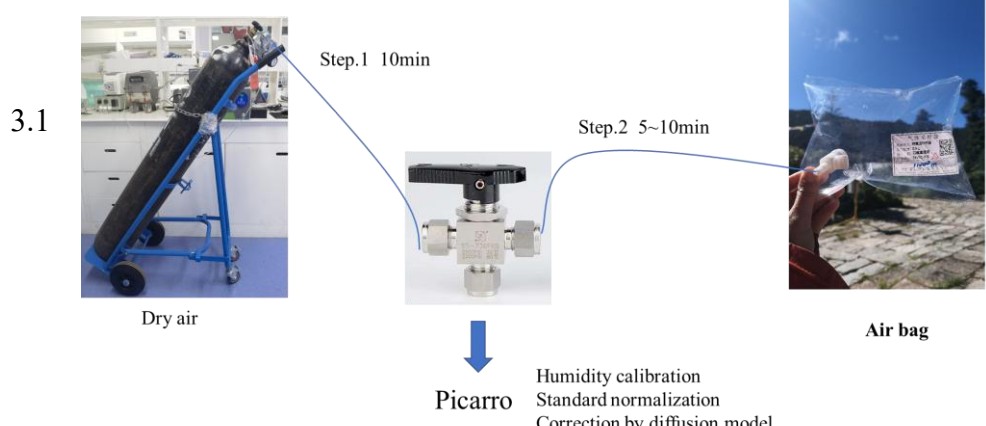

*Figure 2 Setup for isotope measurements using air bags with a Picarro atmospheric water vapor isotope analyzer.*

In this study, we used 0.5 L and 4 L Teflon air bags produced by Dalian Hede Technologies Co., Ltd to collect and store vapor, and measured the vapor isotopes using a Picarro 2130i water isotope analyzer. Based on our testing and the airflow rate set for the Picarro analyzer, the 0.5 L and 4 L bags provided sufficient sample volume for approximately 17 minutes and 130 minutes of analysis, respectively.

Figure 2 illustrates the setup for measuring vapor isotopes in air bags. In this system, the air inlet of the Picarro isotope analyzer was connected to a three-way valve through Teflon or stainless-steel tubing. The other two ports of the three-way valve were attached to the outlet

valve of the air bag and a dry air cylinder, respectively. Sample storage and measurement were conducted in a temperature-regulated room to maintain constant temperature conditions for the air bags and tubing.

In the measurement procedure, we first activated the dry air cylinder and adjusted the pressure reducing valve to 13.8 kPa (2 psi, pounds of force per square inch), within the Picarro water isotope analyzer's recommended range of 13.8–27.6 kPa (2–4 psi) for carrier gas. The instrument's built-in flow regulation maintains a gas flow rate of 40 sccm at 760 Torr, corresponding to ~ 30–50 mL/min in our measurement conditions, ensuring stable sample delivery. We then opened the valve channel connecting the dry air cylinder to the three-way valve, allowing dry air to flow into the isotope analyzer and flush all air pathways for 10 minutes. Subsequently, we closed the dry air channel, opened the air bag outlet valve, and the corresponding valve channel on the three-way valve. This allowed water vapor in the air bags to be analyzed at a constant temperature in the Picarro analyzer, a process lasting between 5 and 10 minutes. Upon completion, we switched the valve to measure dry air. By repeatedly measuring isotopic composition for replicate samples—air samples collected simultaneously under the same conditions—we ensured the consistency of the water vapor measurements.

To correct isotope measurement bias caused by the instrument's sensitivity to different water vapor concentrations(Schmidt et al., 2010), we used the built-in Standard Delivery Module (SDM) of the Picarro L2130-i water vapor isotope analyzer to generate a 500–25,000 ppm water vapor gradient for isotope measurements. We selected 20,000 ppm as a reference humidity level, as this corresponds to the optimal accuracy range of the Picarro analyzer (JingfengLiu et al., 2014; Schmidt et al., 2010). All the measured vapor isotope data were normalized to the VSMOW-SLAP scale using two distinct laboratory reference waters with known isotopic composition. Before conducting daily measurements, we adjusted the quantity of the injected liquid standard to align with the humidity of the external vapor measurements.

## Laboratory permeability experiments

To evaluate the variations of the diffusion of water molecules between the interior and exterior of the air bags, we conducted the following experiments (Table A1), as detailed below:

### 3.2.1    Experiment No. 1: Quantification of $\lambda_{surface}$
3.2

To quantify the water vapor exchange coefficient at the surface, $\lambda_{surface}$, using Eq.10, we filled the empty and clean air bags with dry air and measured humidity variations using a Picarro analyzer at intervals of 1 minute, 2, 4, 6, 8, and 10 hours following the measurement method described in Subsection 3.1. As $\lambda$ is related to the surface area of the sampling bag, measurements were conducted using 0.5L and 4L air bags, with repetitions on both identical and different air bags of the same dimensions (refer to the experiment times in Table 1 and results in Fig.3a). We measured $\lambda_{surface}$ (=k*A/M) under near-surface pressure and temperature we set in a controlled chamber. While k may vary with temperature and pressure, these effects were assumed negligible within the condition of our study. Results will be shown in Subsection 4.1.

### 3.2.2    Experiment No. 2: Quantification of α

To investigate isotope variation patterns and improve measurement accuracy during storage in the air bag to quantify the isotopic fractionation coefficient α, using Eq.13, initial values significantly different from ambient conditions were selected. Empty, clean air bags were first filled with dry air, then sealed by closing the bag valve. To maintain a closed system while injecting reference water, a dedicated injection septum was installed on the valve. After reopening the valve, a fixed amount of laboratory reference liquid water with known isotopic composition was injected into the dry air-filled bag using a 10 µL injection needle. In Experiment No. 2, we ensured that the initial humidity ($q_0$) was approximately equal to the environmental humidity ($q_e$). To ensure $q_0 = q_e$, the environmental vapor concentration was first measured, followed by the calculation and experimental determination of the water volume to be injected into the air bag. Water vapor concentration and isotope variations within the air bags were then measured using a Picarro analyzer at intervals of 5 minutes, 2, 4, 6, 8, and 10 hours. Results will be shown in Subsection 4.1. To ensure data consistency and reliability, we repeated these measurements multiple times using air bags of both the same and different sizes, including 0.5L and 4L bags. For the 0.5L air bags, a separate bag was prepared for each time interval, and water vapor concentration and isotopic compositions were measured once to ensure that the parameter M remained stable without being affected by repeated measurements. However, manual injection of reference water with known isotopic composition introduced some variability, causing minor variations in initial values across all bags. This is evident in Figs 5, 6, and A2, where $q_0 = 1/2\ q_e$ data exhibit slight fluctuations due to the use of separate bags for each time interval. To address this, we repeated the experiment with 4L air bags, measuring the same air bag at different time intervals, which ensured consistent initial conditions at t=0 but allowed M to change over time. When air bags differ only in size, the parameter λ associated with A and M varies, while the isotopic fractionation coefficient α is theoretically constant. Both approaches could contribute to uncertainties in the mismatches between the model and experimental results. Therefore, we incorporated the results from both the 0.5 L and 4 L experiments into our uncertainty estimation of these mismatches, as detailed in Subsections 3.2.3 and 3.4.2.

### 3.2.3    Experiment No. 3: Diffusion model validation

To validate the diffusion model under diverse conditions and evaluate its uncertainties, we repeated Experiment No. 2, but injected different amounts of water with known isotopic composition to achieve a range of humidities from approximately $1/8 * q_e$ to $q_e$. Using the method described in Experiment No. 2, we injected 6 to 50 µL of reference water into a 4L air bag filled with dry air to achieve the desired humidity range. Additionally, we repeated the experiment using two reference waters with distinct isotopic compositions, specifically $\delta^{18}O =$ -58.07‰, $\delta^2H =$ -447.41‰ and $\delta^{18}O =$ -29.84‰, $\delta^2H =$ -222.84‰. To assess extended-duration variations, we also lengthened the time interval to 24 hours.

Once the parameters of the diffusion model have been obtained through Experiments No. 1 and 2, we used this model to simulate the variations in water vapor humidity and isotopic

composition inside the air bag over time for Experiments No. 2 and 3 (refer to Section 2). When simulating these experiments using the diffusion model, we used measurements taken after a 5-minute delay as the initial condition to ensure that it represented complete evaporation of the injected water. We then simulated the temporal variations in humidity and vapor isotopes within the air bag using a 5-minute time step using Eqs. 5 and 8, separately. The resulting outputs (hereafter referred to as 'the diffusion model simulations') will be shown in Subsection 4.2 and code are available as supplementary material.

### Drone-mounted systems and field campaign

We designed and built a collection module for fixed-height sampling, incorporating diaphragm vacuum pumps, a rudder mounted on the drone, and a control module linked to a remote operating system. Before flight, each air bag was evacuated on the ground using the diaphragm pumps. When inactive, these self-sealing pumps effectively isolated the interior of the air bags from the external environment, minimizing diffusion prior to sampling. When the drone reaches a specified altitude, we remotely activate the designated air pump to inflate a specific air bag. Once sampling is complete, the pump is deactivated, and the drone ascends to the next target altitude, where the corresponding air pump inflates another air bag. This process was repeated until all predetermined samples were collected. After each sampling, the pump remained sealed, ensuring no unintended air ingress during descent. Additionally, due to the flexible nature of the air bags, internal and external pressures remained balanced. As air pressure increases during the drone's descent after collection. To further prevent the loss of collected air samples, a one-way valve was installed to block backflow. Additionally, the one-way valve helps prevent large droplets from entering the air bag during the collection process.

The sampling module was mounted on our specially designed high-altitude drones and deployed during a field campaign conducted from June 25, 2020, to October 17, 2020, in the pristine forests of Mountain Laojun, Lijiang, on the southeastern edge of the Tibetan Plateau and the northwestern of the Yunnan-Guizhou Plateau, China. We collected water vapor samples every 500 meters, starting from near the surface along the vertical profile. To optimize sampling across different altitude ranges, we deployed UAVs designed for varying flight altitudes. Generally, the UAV operating at lower altitudes collected samples at seven heights from 4,000 to 7,000 meters in a single flight. The mid-altitude UAV collected samples at four heights from 7,500 to 9,000 meters in one flight, while the high-altitude UAV collected samples at four heights from 9,500 to 11,000 meters in two flights. Each flight took approximately 20~30 minutes. In case of any disruptions during sampling, we repeated the process until a complete vertical profile was obtained. At the beginning of the experiment, we also collected replicate samples at each height to ensure data consistency.

By integrating high-altitude drone sampling with subsequent water vapor isotope analysis using the Picarro analyzer at the surface, we obtained vapor isotopic profiles up to an altitude of 11 km. Unlike conventional methods, such as cryogenic vapor sampling, this approach requires much less sample volume and allows for timely measurements. Additionally, compared to large aircraft, airships, and balloon-based observations, this method is relatively low-cost and supports more flexible and long-term observations.

### 3.4 Application of diffusion model to vertical profiles

#### 3.4.1 Estimating the air mass in the bag

$\lambda$ is defined as $k*A/M$ and depends on the air mass M in the bag. In drone-based vertical sampling, M varies with altitude due to pressure changes, requiring an estimate of $\lambda$ for different altitudes ($\lambda_{alt}$). However, since $\lambda$ is an intrinsic property of the bag material, its apparent variation reflects uncertainties in estimating collected M, which depend on atmospheric pressure (P), sampling time, and pump efficiency ($\varepsilon$):

$$M_{alt} = M_{surface} * \frac{P_{surface}}{P_{alt}} * \frac{Sampling\ time_{alt}}{Sampling\ time_{surface}} * \varepsilon \qquad (15)$$

where $M_{alt}$ is the air mass collected at a different altitude and $M_{surface}$ represents the air mass collected at the surface. At higher altitudes, where the air pressure ($P_{alt}$) is lower than at the surface ($P_{surface}$), less air will be pumped into the air bag. To compensate for this effect, a longer sampling time was used to collect air at higher altitudes (Sampling time$_{alt}$) than at the surface (Sampling time$_{surface}$) (Fig.A1).

Given that $\lambda_{alt}$ is proportional to $M_{alt}$, we calculated it as:

$$\lambda_{alt} = \lambda_{surface} * \frac{P_{surface}}{P_{alt}} * \frac{Sampling\ time_{alt}}{Sampling\ time_{surface}} * \varepsilon \qquad (16)$$

where $\lambda_{surface}$ is the $\lambda$ quantified experimentally at the surface.

Since air pressure and sampling times were directly measured, the primary source of error for $M_{alt}$, and consequently $\lambda_{alt}$, arises from pump efficiency ($\varepsilon$), which may decrease over time and at lower pressures. Using the estimated $\lambda_{alt}$, the observed vertical isotope profiles were corrected based on Eq.14 from Section 2.2. The uncertainty estimation is discussed in Section 3.4.2.

#### 3.4.2 Method of uncertainty estimation

Potential errors in correcting vertical profiles using the diffusion model include estimates of $\lambda_{surface}$, $\alpha$, pump efficiency ($\varepsilon$), and mismatches between model and experiments (Table 1). We detail each below:

1) $\lambda_{surface}$ uncertainty: laboratory experiments provided upper and lower bounds on $\lambda_{surface}$ (Subsection 4.1). These were used for error estimation.

2) $\alpha$ uncertainty: the $\lambda$ and $\lambda/\alpha$ were first estimated from several experiments, from which $\alpha$ was calculated. Their averaged values were used separately to parameterize the model. As highlighted in Subsection 3.2.2, estimating $\lambda/\alpha$ (and subsequently calculating $\alpha$) requires results from cases where $q_0$ equals $q_e$, minor variations in $q_0$ and fluctuations in $q_e$ could introduce non-systematic discrepancies between the model and experimental results. Consequently, for analyzing the contribution of $\alpha$ to uncertainties, only $\alpha$ derived from experiments where the model closely matched the majority of experimental results were considered. Selection criteria for these experiments included minimal deviation between $q_0$ and $q_e$, minimal deviation between experimental data and simulations, and stable $q_e$, ensuring the reliability of the chosen $\alpha$.

3) Pump efficiency ($\varepsilon$) uncertainty: The efficiency of the pump may decline over time or
vary with atmospheric pressure, affecting the collected air mass M. To account for this, we
applied a conservative uncertainty range of 0.75 to 1.25 relative to surface conditions, ensuring
the full range of possible variations in $M_{alt}$ was considered.
4) Model-experiment mismatches: We compared model simulations with experimental data
across 87 cases, calculating the average absolute discrepancy. These mismatches were included
as an additional uncertainty component.
Total Uncertainty Calculation: The maximum discrepancy across all calibration results—
using the full uncertainty range for $\lambda_{surface}$, $\alpha$, and pump efficiency ($\varepsilon$)—was determined. The
model-experiment mismatch was then added as an independent error component. The final
uncertainty estimates, reported in Subsections 4.3 to 4.5, account for all potential error.

*Table 1 Uncertainties and estimation methods*

| Uncertainties | Number of experiments | Estimation method | Used value (min~max for error estimation) |
|---|---|---|---|
| $\lambda_{surface}$ | 7 (0.5L bags); 6 (4L bags) | Obtained from lab experiments (max & min) | For 0.5L air bags: 0.031(0.0291~0.0317); For 4L air bags: 0.0255(0.0250~0.0259) |
| $\alpha$ | 4 | Obtained from lab experiments | $\alpha^{18}O$ is 1.0241, $\alpha^2H$ is 1.0451 ($\alpha^{18}O$ is 1.0254, $\alpha^2H$ is 1.0506 and $\alpha^{18}O$ is 1.0264, $\alpha^2H$ is 1.0380) |
| Pump efficiency ($\varepsilon$) | - | 0.75~1.25 relative to surface conditions | 0.75~1.25 |
| Mismatches between model and experiments | 87 | Average of all differences between experimental data and simulations | 0.5 ‰ for $\delta^{18}O$, 4.1 ‰ for $\delta^2H$, 2.9 ‰ for d-excess |

**Satellite isotope data from IASI**

Several satellite missions have contributed to water vapor isotope observations, including the Tropospheric Emission Spectrometer (TES) onboard Aura (2004–2018) (Worden et al., 2006), the Scanning Imaging Absorption Spectrometer for Atmospheric Cartography (SCIAMACHY) onboard Envisat (2002–2012), the Atmospheric Infrared Sounder (AIRS) onboard Aqua (since 2002) (Worden et al., 2019), and the Tropospheric Monitoring Instrument (TROPOMI) onboard Sentinel 5 Precursor (since 2017)(Schneider et al., 2020). In this study, we use the MUSICA retrievals from the Infrared Atmospheric Sounding Interferometer (IASI) onboard METOP due to its broad spatiotemporal coverage, vertical profiling capability, and the availability and accessibility of its dataset (Diekmann et al., 2021). It has a horizontal footprint of approximately 12 km at nadir (directly below the satellite), increasing with the angle of observation. This configuration ensures nearly global coverage twice daily.

In this study, as no satellite retrievals of $\delta^{18}O$ are currently available, we compared our observed vapor $\delta^2H$ profiles up to the upper troposphere with satellite observations. Due to the intermittent availability of IASI data at any given location, we limited our comparison of observational results across various altitudes at our study site to days when IASI data were available. Satellite measurements, particularly for vertical profiles of water vapor isotopes, are inherently different from direct sampling, they represent a vertical average over layers determined by the averaging kernels (Rodgers and Connor, 2003; Worden et al., 2006). Therefore, their comparability with ground-based or drone-based observations, which provide high-resolution local data, is limited. The MUSICA retrievals from the IASI satellite instrument provides water vapor isotope data at three altitude levels: 1-3 km in the lower troposphere, 4-7 km in the mid-troposphere, and 8-12 km in the upper troposphere. Given that our study started at an altitude of 3856 m, we used the retrieved $\delta^2H$ data for the 4–7 km and 8–12 km levels. However, these measurements represent a vertical average over layers determined by the

averaging kernels (Rodgers and Connor, 2003; Worden et al., 2006). While using averaging
kernels to smooth the observed profile could facilitate a more quantitative analysis, the multi-
level data with averaging kernels for 2020 are not publicly available, we therefore simply
averaged the observations for the corresponding altitudes. Consequently, the comparison
remains mainly qualitative.

## 4 Results

### Parameter estimates

The $\lambda_{\text{surface}}$ was determined through laboratory Experiment No. 1. Based on Eq.10, the
parameter $\lambda_{\text{surface}}$ was estimated to be 0.0312 (uncertainty range: 0.0291 to 0.0317) for the 0.5
L air bags and 0.0255 (uncertainty range: 0.0250 to 0.0259) for the 4 L air bags (Fig. 3a, Table
1).
The isotopic fractionation coefficients $\alpha$, were determined through laboratory Experiment
No. 2. From the averaged results of these measurements, using Eq.13, the water vapor isotopic
exchange coefficient, expressed as $\lambda/\alpha$, was estimated to be 0.0249 for $^{18}O$ and 0.0244 for $^{2}H$
(Fig. 3b and c). Consequently, $\alpha^{18}O$ was estimated to be 1.0241 (0.0255/0.0249), and $\alpha^{2}H$ was
estimated to be 1.0451 (0.0255/0.0244) (Fig. 3b and d). Two additional sets of fractionation
coefficients $\alpha$, were obtained: $\alpha^{18}O = 1.0254$, $\alpha^{2}H = 1.0506$, and $\alpha^{18}O = 1.0264$, $\alpha^{2}H = 1.0380$
(Table 1).
The specific parameter values obtained in this study pertain to the Teflon air bags used in
the aforementioned tests, conducted at an ambient temperature of 16°C. These values depend
on bag material, temperature, and pressure, which should be considered when applying the
model under different conditions. We also noted batch-to-batch variations among air bags from
the same manufacturer. We apply $\alpha$ measured under ground-level storage and measurement
conditions, assuming negligible temperature and pressure effects during the short (10–20 min)
drone-based sampling period. Future work is needed to quantify these dependencies.

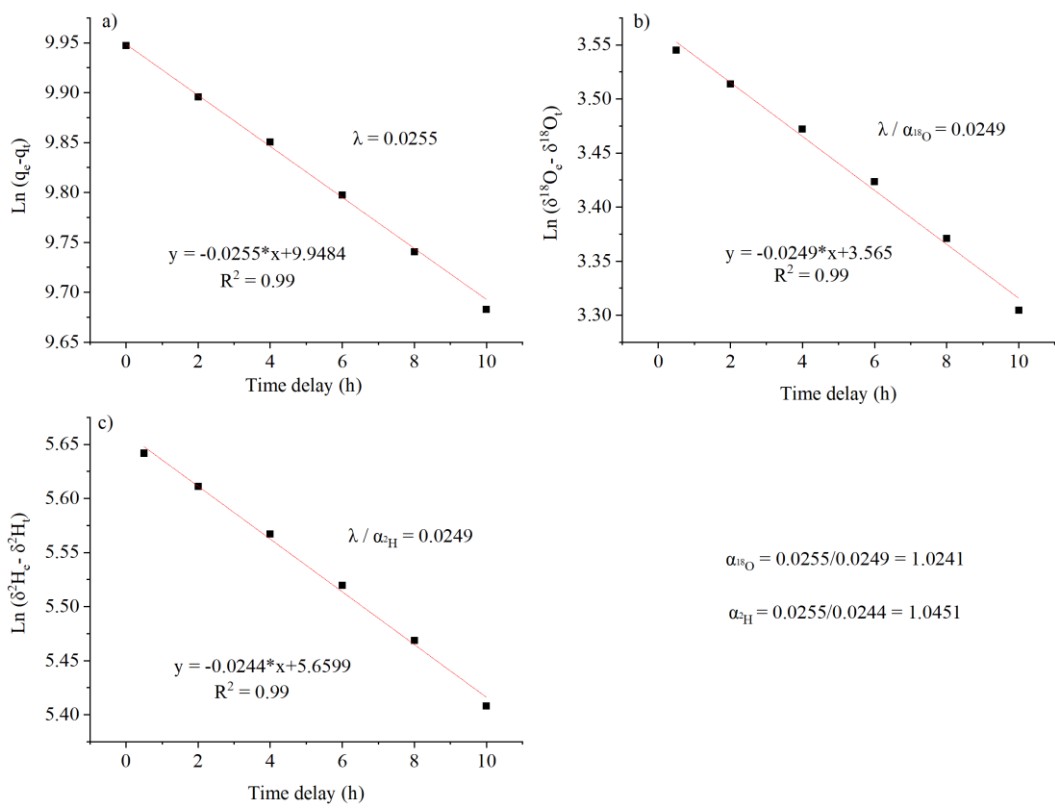


*Figure 3 Determination of 3 parameters of the diffusion model : λ_surface (a) from Experiment No. 1, α¹⁸O (b), and α²H (c) from Experiment No. 2.*

**Diffusion model validation**

**4.2.1    General case**

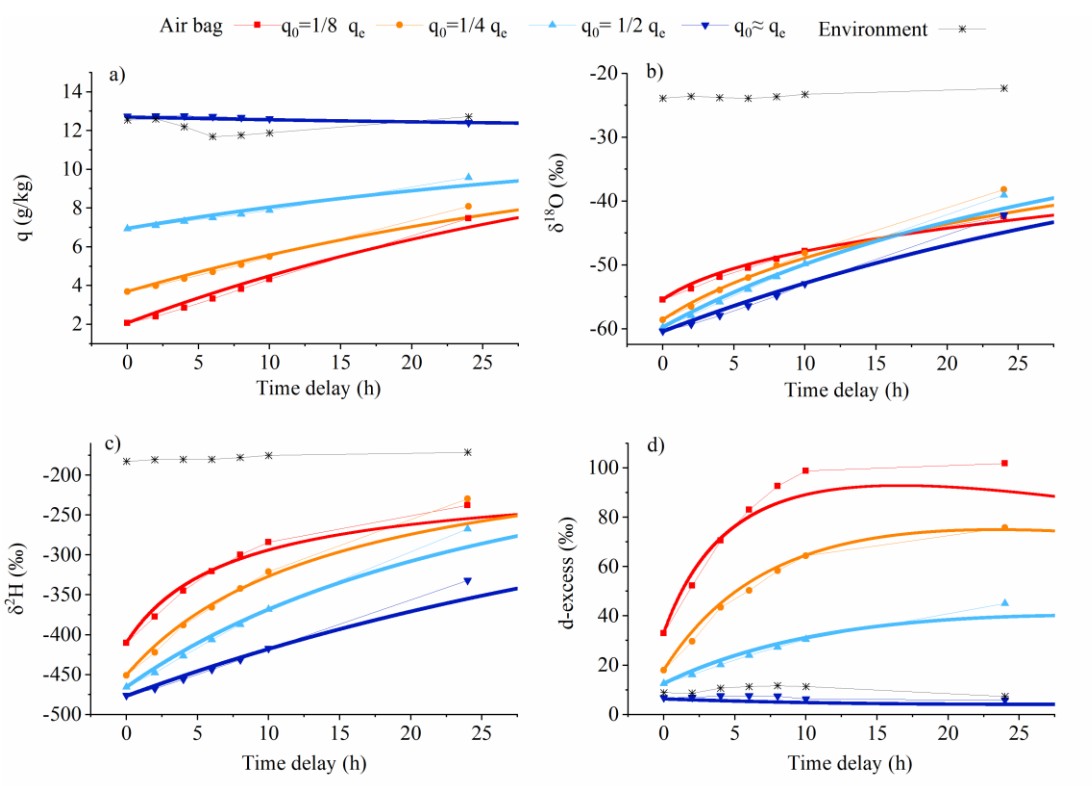

*Figure 4: Comparison of the variation within the air bag over time in laboratory permeability experiments (markers) and diffusion model simulations (lines) across a range of initial humidities at t = 0 ($q_0$), ranging from approximately 1/8 \* $q_e$ to $q_e$ ($q_e$ denotes the environmental humidity), under the condition that the water vapor isotopic composition in the air bag significantly differs from ambient values : a) humidity; b) $\delta^{18}O$; c) $\delta^2H$, d) d-excess.*

To validate the model, we used Experiment No.3 described in Subsection 3.2. The
diffusion model simulations (lines in Figs. 4, 5, 6, and Fig. A2) are in close agreement with our
experimental observations (markers in Figs. 4, 5, 6, and Fig. A2), showing consistency in
humidity, $\delta^{18}O$, $\delta^2H$, and d-excess variations, with only minor deviations. Shorter storage times
produce fewer deviations. When the humidity inside the air bag is lower than the ambient level,
vapor from the environment enters the air bag, resulting in a gradual increase in humidity (Fig.
4a and Fig. A2). Meanwhile, because the water vapor isotopic composition in the air bag is
significantly lower than the ambient values, $\delta^{18}O$ and $\delta^2H$ in the air bag gradually increase and
toward ambient values as ambient moisture enters the bags over time (Fig. 4b and c). In contrast,
the d-excess increase as the time delay progresses due to kinetic fractionation during moisture
diffusion into the air bag (Fig. 4d), as detailed in the following Subsection 4.2.2.
We observed that some curves unexpectedly intersect, which can be understood by

analyzing Eq. 8. The first term $(\frac{\lambda}{\alpha} * (R_e - R_{(t)})$ continuously drives $R_{(t)}$ towards $R_e$, while the
second term $(\frac{\lambda}{q_{(t)}} * (q_e - q_{(t)}) * (\frac{Re}{\alpha} - R_{(t)}))$ modulates the rate of change. Initially, for simulations
with lower $q_0$ (e.g., the red and orange curves), $q_e - q_{(t)}$ is large, making the second term
significant and positive, thereby increasing $R_{(t)}$ more rapidly. However, as $R_{(t)}$ exceeds $R_e/\alpha$, the
sign of this term reverses, slowing down the increase in $R_{(t)}$ compared to other curves. In
contrast, curves with higher initial $q_0$ (e.g., blue curve) experience a steadier growth and
eventually surpass the initially faster-growing curves, leading to the observed crossing.
### 4.2.2    Particular cases

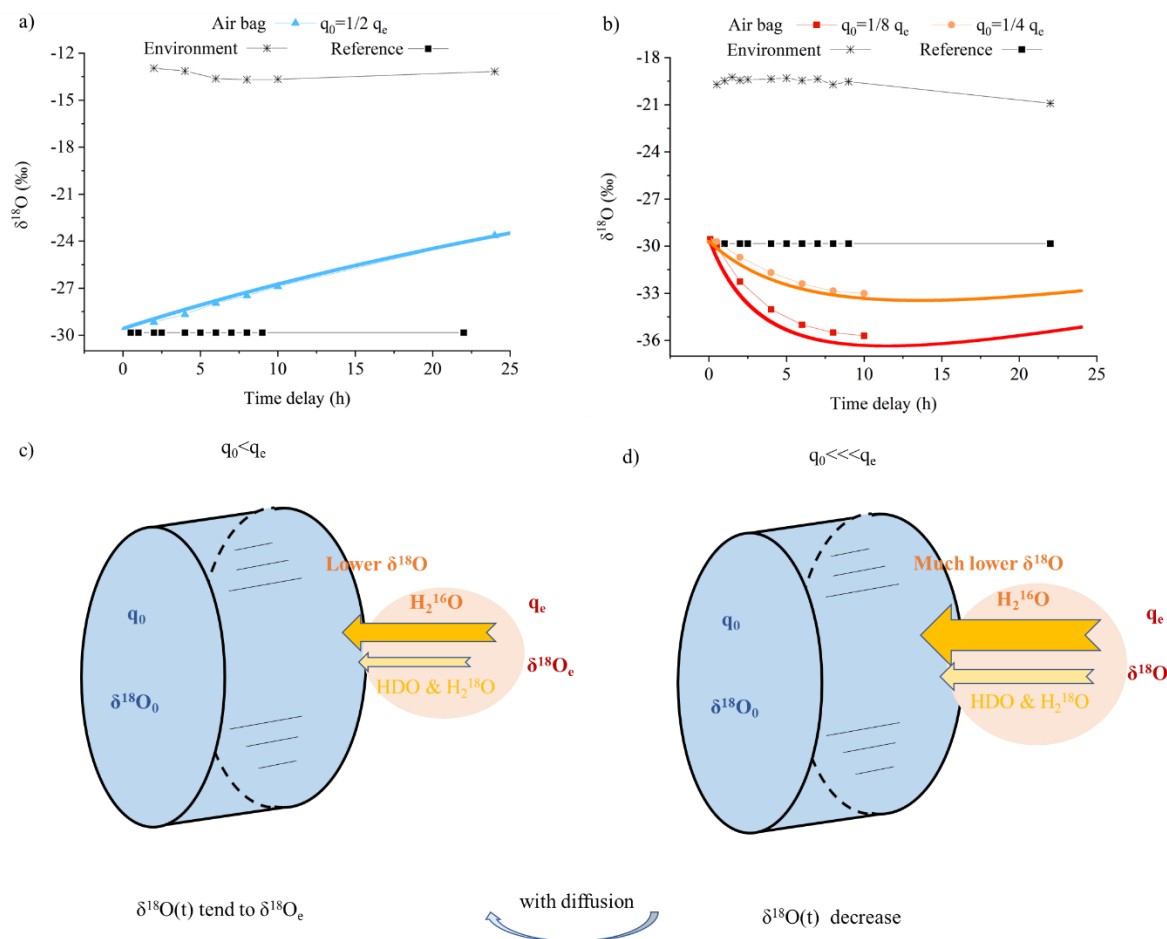


*Figure 5    (a-b) Variations of $\delta^{18}O$ under different conditions: (a) when both the differences between internal ($\delta^{18}O_0$) and external ($\delta^{18}O_e$) $\delta^{18}O$ as well as between internal ($q_0$) and external ($q_e$) humidity are not significant, $\delta^{18}O$ gradually increases toward equilibrium; (b) when $q_0$ is significantly lower than $q_e$, a stronger vapor influx causes enhanced kinetic fractionation, leading to a decrease in $\delta^{18}O$. (c-d) Corresponding schematics: (c) illustrates the mechanism for (a), where a weaker humidity gradient results in slower isotopic shifts, while (d) corresponds to (b), showing intensified fractionation with a larger gradient, with arrows indicating vapor flux direction and fractionation intensity. $\delta^{18}O$ (t) is the variation of $\delta^{18}O$ within the air bag over time. In (a) and (b), the colored lines represent model simulations based on Eq.8, using parameterization from Experiments No. 1 and 2. Colored square markers show the corresponding experimental observations within the air bags. Black square markers*

*indicate the initial values inside the air bag, which remain constant over time and serve as a reference for comparison (legend: Reference).*

Due to isotopic kinetic fractionation, lighter $H_2^{16}O$ molecules preferentially diffuse into

the air bag compared to heavier isotopologues such as $HD^{16}O$ and $H_2^{18}O$, resulting in a vapor
flux with lower $\delta^{18}O$ relative to ambient vapor (Fig. 2). Moreover, variations within the air bag
are driven by differences in water vapor content and isotopic ratios between its interior and
exterior, as described in the diffusion model in Section 2. As shown in the initial experiment
(Fig. 4b and c), when the internal $\delta^{18}O$ and $\delta^2H$ are significantly below the ambient values, the
$\delta$-values of the diffusing vapor, although lower than ambient, still exceeds the initial internal $\delta$-
values, leading to a gradual increase towards the ambient values; diffusion simply equilibrates
the isotopic composition in the bag and the environment. When the disparity between internal
and external $\delta^{18}O$ or $\delta^2H$ is not very substantial, and humidity differences are also minimal, the
weaker diffusive gradient produces less net kinetic fractionation. This results in a small amount
of vapor with lower $\delta^{18}O$ and $\delta^2H$ than the ambient moisture entering preferentially, but not
falling below internal initial values, thereby still drives a progressive increase in the internal
values towards ambient moisture (Fig. 5a and c). In contrast, with the same initial contrast
between internal and ambient $\delta$-values, that is, the disparity between internal and external $\delta^{18}O$
and $\delta^2H$ is less pronounced, but $q_0$ is much lower than $q_e$, there is a stronger net flux into the
bag, and this flux fractionates more rapidly; much more vapor with significantly lower $\delta^{18}O$
and $\delta^2H$ than the ambient moisture (and lower values than the initial internal vapor) enters the
air bags and dominates their isotopic composition, thereby reducing the internal $\delta$-values
(Fig.5b and d). The smaller the difference in humidity and isotopic composition between the
inside and outside of the air bag, the slower and smaller the isotopic change in the vapor within
the air bag (Fig. 5 and Fig. 6).

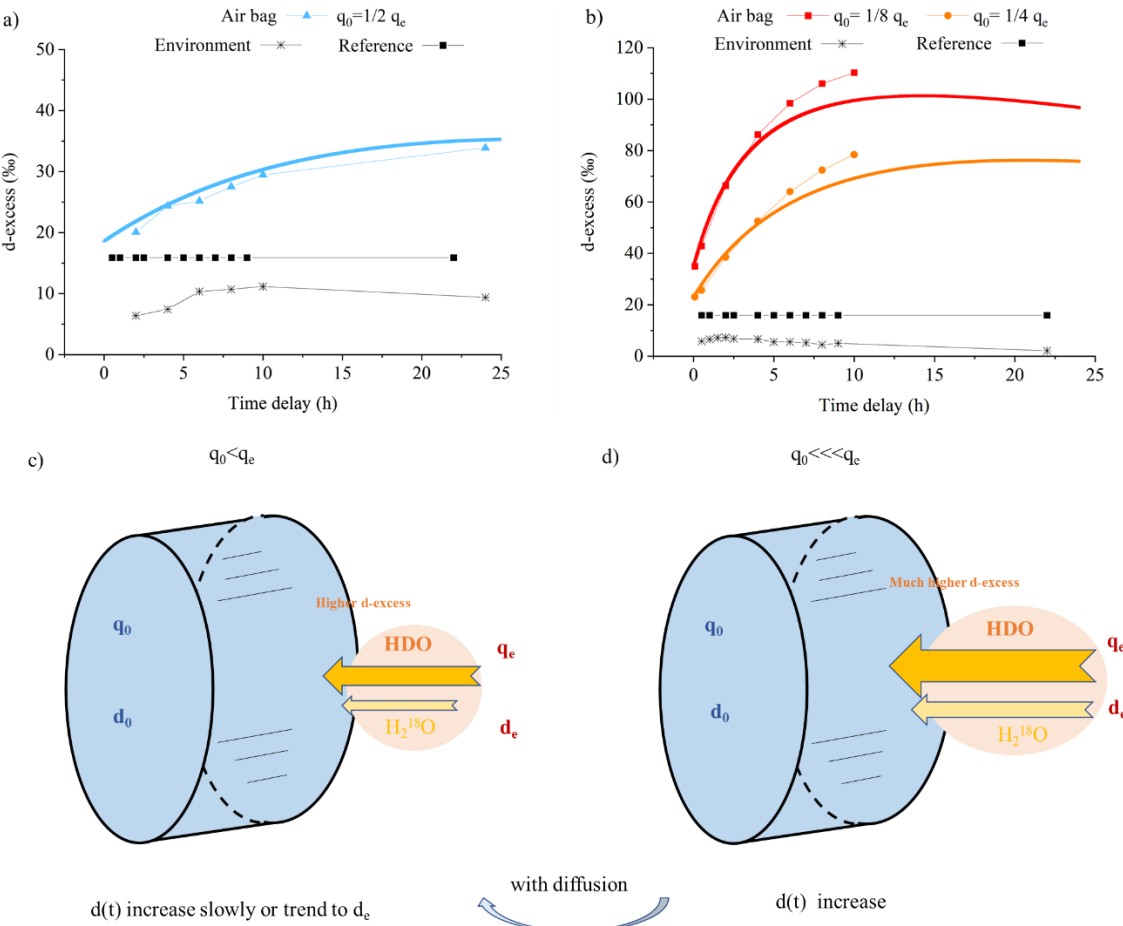

*Figure 6 (a-b) Same as Fig.5, but showing the evolution of d-excess: (a) when the difference*
*between the humidity inside ($q_0$) and outside ($q_e$) the air bag is not significant, d-excess*
*increases gradually; (b) when $q_0$ is significantly lower than $q_e$, a stronger vapor influx*
*enhances kinetic fractionation, causing a more rapid d-excess increase. (c-d) Corresponding*
*schematics: (c) illustrates the mechanism for (a), where a smaller humidity gradient results in*
*slower isotopic shifts, while (d) corresponds to (b), showing intensified fractionation with a*
*larger gradient. $d_0$ indicates the initial d-excess at t = 0, $d_e$ represents the d-excess in the*
*environment. d(t) denotes the variation of d-excess within the air bag over time.*

In addition, because HDO and $H_2^{18}O$ diffuse at similar rates the magnitude of the kinetic
fractionation for D and $^{18}O$ is similar. However, since d-excess reflects deviations relative to
the 8:1 fractionation ratio typical of equilibrium processes, the tendency is for kinetic
fractionation during diffusion to contribute vapor with high d-excess and cause an increase in
d-excess during air bag storage as water vapor is added to the bag (Fig. 6a and c). When the
humidity difference between the inside and outside of the air bag increases, the d-excess of the
incoming vapor flux increases as a result of more intensive kinetic fractionation. This leads to
a faster increase in the vapor d-excess inside the air bag (Fig. 6b and d).

Regardless of the differences in water vapor humidity and isotopic compositions inside
and outside the air bag, the diffusion model simulations closely match the experimental
observations (Figs.4, 5 and 6). Using the method described in Subsection 3.4.2, the average
difference between all simulations and experimental data for each parameter represented the

model-experiment mismatch: 0.5 ‰ for $\delta^{18}O$, 4.1 ‰ for $\delta^2H$, and 2.9 ‰ for d-excess.
**Raw and corrected vertical profiles**
Here, we present a summary of drone-based observations from the field campaign at Mount
Laojun, Lijiang, on the southeastern edge of the Tibetan Plateau and the northwestern of the
Yunnan-Guizhou Plateau, China, conducted between June 25, 2020, and October 17, 2020.

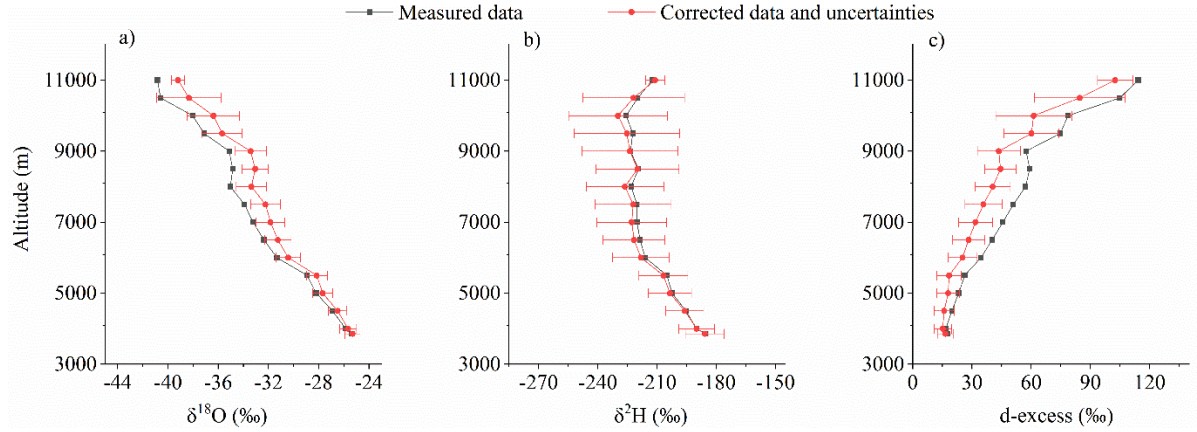


*Figure 7 Comparison of vertical profiles for the mean values of all raw measurements and corrected data from June to October, and associated uncertainties for $\delta^{18}O$ (a), $\delta^2H$ (b) and d-excess (c).*

As altitude increases, vapor $\delta^{18}O$ and $\delta^2H$ decrease due to condensation and precipitation
processes that occur as air masses ascend, which preferentially remove heavier isotopes
following Rayleigh distillation (Dansgaard, 1964). Meanwhile, the d-excess rises. This pattern
aligns with previous observations in the lower troposphere (He and Smith, 1999; Salmon et al.,
2019) and simulations of complete vertical profiles (Bony et al., 2008).
In our observations, the variation in $\delta^{18}O$ across the vertical profile from ground level at
3856 meters to 11 km is approximately 10-15‰. However, as altitude increases, the air becomes
progressively drier, leading to a greater disparity in humidity between the air collected in the
air bag and the surface storage environment. This aligns with the variations in Subsection 4.2.2
(Figs. 5b, 5d, and 6b, 6d). The strong kinetic fractionation driven by the diffusion of air into the
air bag results in a decrease in the water vapor $\delta^{18}O$ within the bag. After applying model
corrections, the corrected $\delta^{18}O$ inside the bag increased slightly compared to pre-correction
levels. As described in Subsection 4.2, vapor flux with higher d-excess entering the bag
increases the d-excess inside. As a compensation, the diffusion model applies corrections,
resulting in a reduced d-excess after correction (Fig. 7c and 10).
**Uncertainty estimates**
According to the method of uncertainty estimation elaborated in Subsection 3.4.2, we
investigated four errors sources to evaluate uncertainty (Table 1).

       4.4                                    21

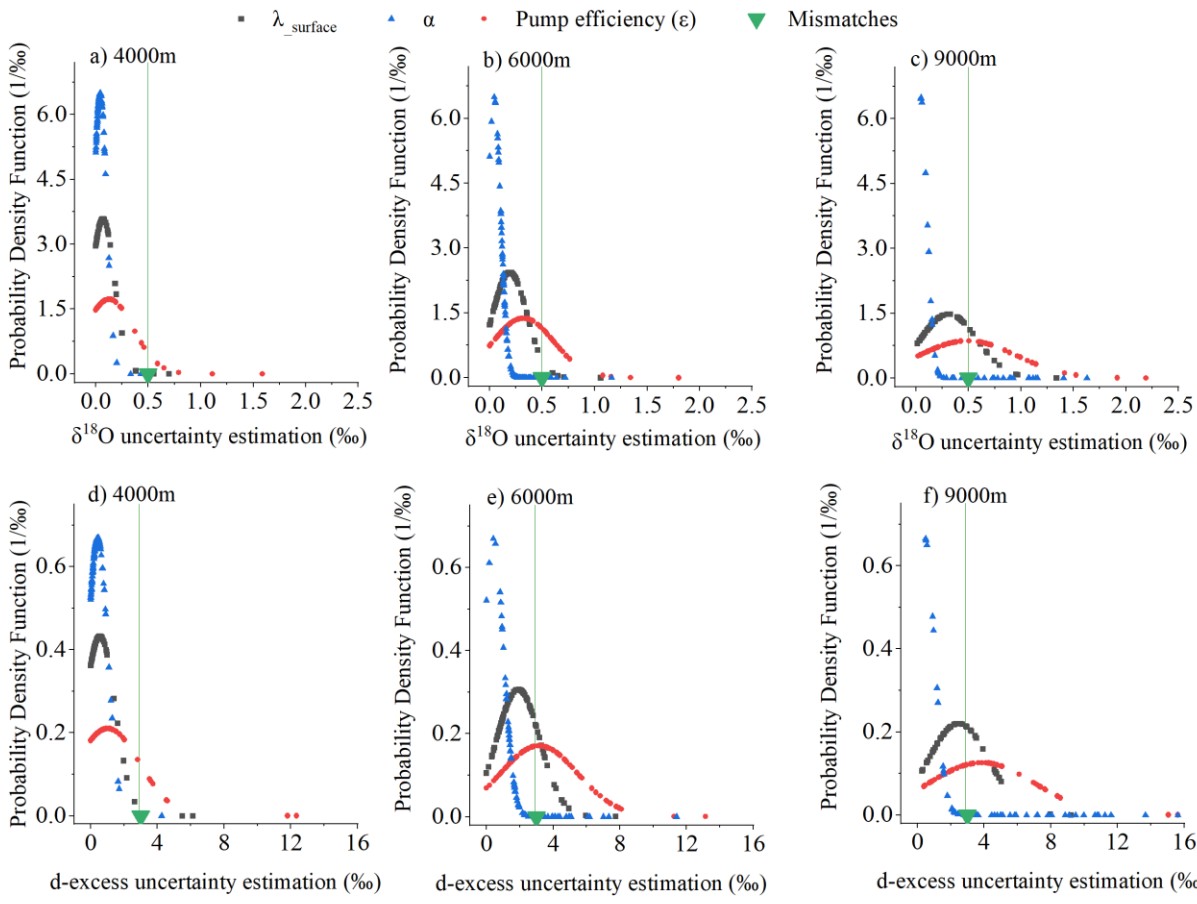

*Figure 8: Contributions of different sources of uncertainty for $\delta^{18}O$ (a, b, c for 4000m, 6000m, 9000m, respectively) and d-excess (d, e, f for 4000m, 6000m, 9000m, respectively). The green triangle shows the average mismatch between the model and experimental results. The peak position of each probability density function (PDF) represents the most likely uncertainty value for a given source, while the width of the distribution reflects the variability.*

We represent the contribution of each source of uncertainty to vertical vapor $\delta^{18}O$ and d-
excess measurements at different altitudes using probability density function plots (Fig. 8).
Each PDF peak indicates the most probable uncertainty value, while its width reflects
sensitivity to that parameter. Narrow PDFs suggest stable, well-constrained uncertainty sources,
whereas broader PDFs indicate greater variability and a more diffuse impact on the correction
outcome. The mismatch between the model and the experiment (green marker), calculated
using the method described in Subsection 3.4.2, is assumed to remain constant across all
altitudes. The other three error sources manifest as unimodal normal distributions at various
altitudes for both $\delta^{18}O$ and d-excess. Errors due to uncertainties in pump efficiency ($\varepsilon$) are the
main source, exhibiting the largest spread (Fig. 8) and increasing with altitude (Fig. 9). Errors
derived from $\lambda_{surface}$ and $\alpha$ also increase with altitude (Figs. 8 and 9). As a result, at higher
elevations, the satellite $\delta^2H$ differs more from the measured and corrected data than at lower
altitudes (Fig. 10). This pattern arises because $\lambda_{alt}$ deviates more from $\lambda_{surface}$ at higher elevations
(Eq.16), primarily due to increased errors in estimating $M_{alt}$, amplifying correction errors.
Moreover, the humidity and isotopic disparity between the air captured in the air bag and lower-
altitude ambient air widens with altitude, requiring more intensive corrections. Consequently,
both the uncertainty (Figs. 8 and 9) and the magnitude of the diffusion correction (Fig. 7)
increase with altitude.

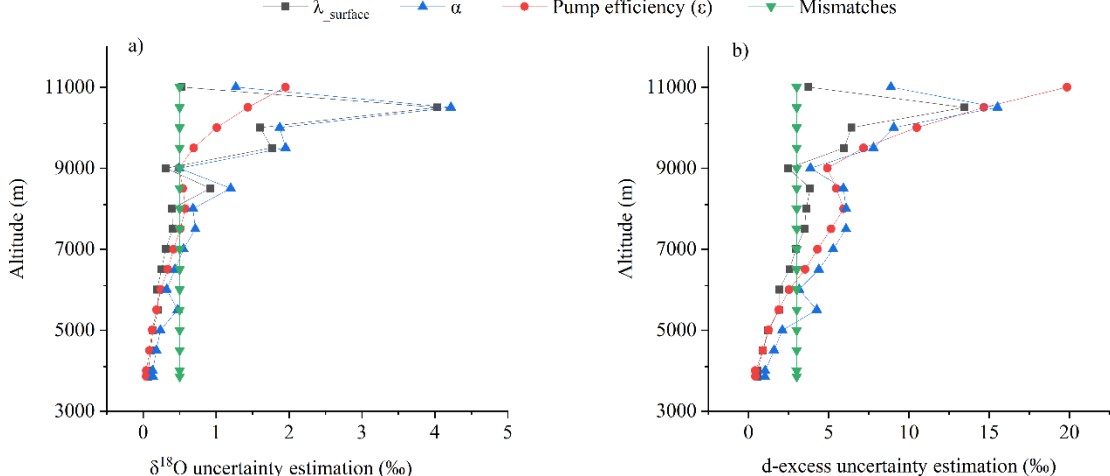

*Figure 9 Mean uncertainty of $\delta^{18}O$ (a) and d-excess (b) with altitude for different sources.*

On the contrary, errors pertaining to $\lambda_{surface}$ and α diminish at altitudes exceeding 10,500
meters. This phenomenon may be attributed to our sampling methodology. Although we
collected air samples in sequence from low to high altitudes, these samples were measured in
reverse order, from high to low altitudes. Consequently, samples taken from the highest
altitudes had the shortest storage durations, typically ranging from 10 minutes to two hours.
Additionally, we extended the sampling times with increasing altitude (Fig. A1). These
extended sampling periods and reduced storage durations help to partially offset the amplified
disparities observed between the raw and corrected profiles at higher altitudes. During the
drone-based sampling period, which lasted around 10-20 minutes, the external conditions
impacting the air bags were not the ground-level environmental values used in the model but
rather those of the vertical profile. However, the humidity at higher altitudes is lower than at
ground level, and the isotopic composition are closer to those inside the airbag. Consequently,
its impact on the airbag's internal conditions is less significant than suggested by using ground-
level environmental values. Therefore, the overestimated error in our model accounted for these
potential discrepancies.
The combined uncertainty from all sources, including $\lambda_{surface}$, α, pump efficiency (ε), and
model-experiment mismatches, result in a total uncertainty of approximately 1‰ for $\delta^{18}O$ and
8‰ for d-excess across 98% of the data. The corresponding 95% confidence intervals are
approximately 0.9‰ for $\delta^{18}O$ and 6.9‰ for d-excess. This range is acceptable when compared
to the substantial vertical variations we observed (~20‰ for $\delta^{18}O$ and ~100 ‰ for d-excess).
Among these sources, ε contributes the largest uncertainty, particularly at higher altitudes (Fig.
8), likely due to the conservative uncertainty range we applied to account for potential
reductions in collected air mass at high altitudes. Additionally, $\lambda_{surface}$ and α also contributes
considerably to the total uncertainty (Figs. 8 and 9). To mitigate this, we recommend conducting
multiple measurements to obtain an averaged value and performing repeated parameter
validation to ensure robustness.

**Comparison with Picarro measurements and satellite data**

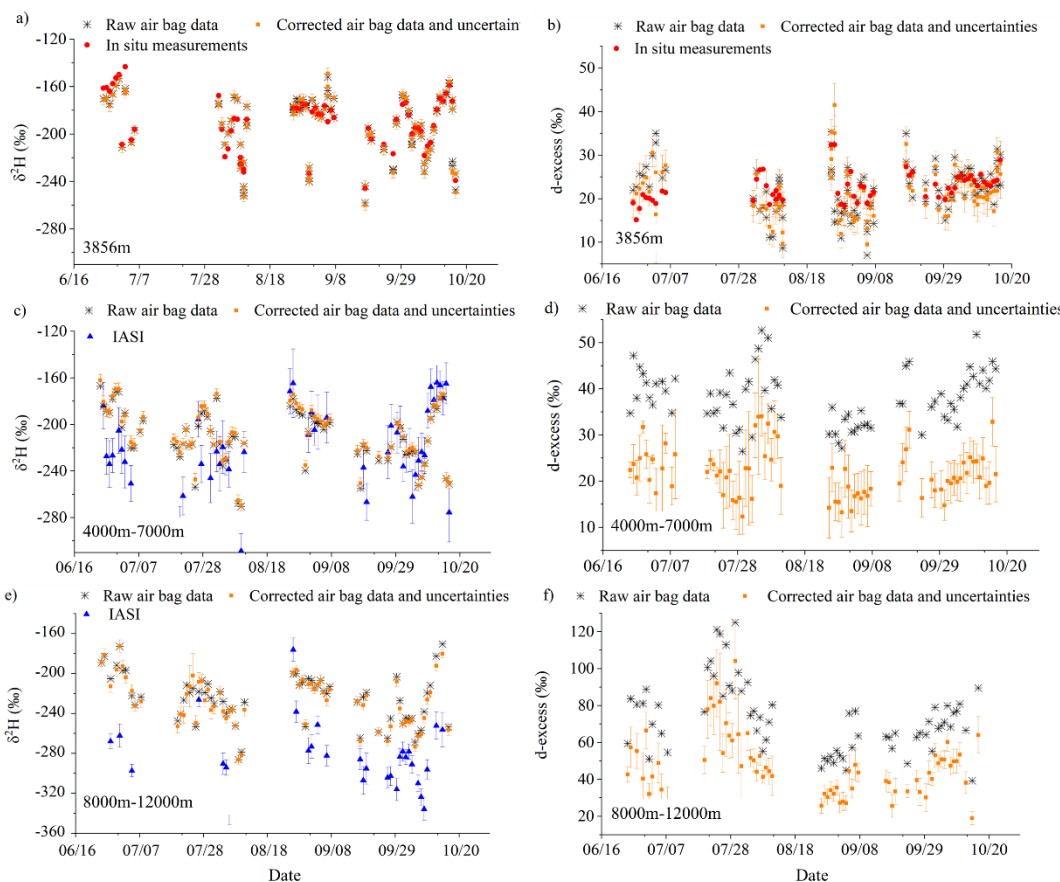


*Figure 10 Time series comparison for δ²H (a, c, e) and d-excess (b, d, f):*

*(a, b) Raw and corrected (with uncertainties) instantaneous air bag measurements at the surface (3856 m), compared with Picarro in-situ measurements at the same altitude, averaged over the sampling period.*

*(c, d) Raw and corrected (with uncertainties) altitude-averaged air bag measurements from 4000 m to 7000 m, compared with satellite data (IASI), which also include uncertainty ranges accounting only for retrieval-fit noise and the atmospheric temperature a priori constraint.*

*(e, f) Raw and corrected (with uncertainties) altitude-averaged air bag measurements from 8000 m to 12000 m, compared with satellite data (IASI).*

The left panel of Fig.10 (Figs. 10 a, c, and e) shows the comparison of raw and corrected
water vapor δ²H measurements at different altitudes with in-situ surface-level measurements
on the Picarro or IASI satellite data at corresponding altitudes.
At near-ground level, both the raw and corrected δ²H and d-excess values from the bag
measurements show close agreement with the in-situ Picarro observations (Fig. 10a and b),
indicating minimal diffusion effects under nearly identical humidity and isotopic conditions
between the inside and outside of the bags. Most data points are distributed along the 1:1 line
(Fig. A3), while d-excess shows slightly larger scatter. After applying the correction, the

agreement improved: the correlation coefficient between the bag and Picarro $\delta^2H$ values increased from 0.90 to 0.99, and for d-excess from 0.41 to 0.72. The mean absolute error (MAE) between the corrected bag and Picarro measurements is 3.0‰ for $\delta^2H$ and 1.4‰ for d-excess. These discrepancies are within the uncertainty range derived from laboratory diffusion experiments and model–experiment mismatches, which were comprehensively incorporated in the error estimation (Sections 3.2 and 3.4): 0.5 ‰ for $\delta^{18}O$, 4.1 ‰ for $\delta^2H$, and 2.9 ‰ for d-excess. The differences also can be attributed to the fact that the Picarro in-situ observations are period-averaged, whereas the bag samples represent instantaneous values.

  To compare corrected measurements with observations at higher altitudes, we refer to the IASI satellite dataset. We acknowledge that this comparison is qualitative, as satellite data represent vertical averages over layers defined by averaging kernels, and therefore differ in measurement footprints (both horizontal and vertical) and spatio-temporal sampling disparities (Shi et al., 2020). A more quantitative analysis could be facilitated if an averaging kernel is used to smooth the observed profiles (Herman et al., 2014), but even after such adjustment, substantial differences would likely persist. IASI provides only two overpasses per day (around 09:30 LT AM and PM), whereas our drone-based measurements were conducted locally during daytime. The IASI footprint (~12 km at nadir) provides coarse spatial averaging, whereas our measurements are local and instantaneous, limiting its ability to capture fine-scale variability at the sampling site. Importantly, IASI cannot observe through or below thick clouds such as anvils, and thus mainly samples air outside convective detrainment regions. Because convective detrainment isotopically enriches vapor (Kuang et al., 2003; Moyer et al., 1996; Smith et al., 2006; Vries et al., 2021; Webster and Heymsfield, 2003), the air masses observed by IASI— being farther from detrainment—are expected to be more depleted. This cloud-sampling limitation may therefore contribute to the lower $\delta^2H$ values retrieved by IASI in the upper troposphere compared with our in-situ observations. Such fundamental mismatches in vertical weighting, temporal coverage, and spatial representativeness are noted in previous satellite–in-situ intercomparisons, which found that satellites capture large-scale isotopic gradients but smooth out fine-scale vertical and diurnal variations (Herman et al., 2014; Lacour et al., 2018). These complementary characteristics highlight the value of combining satellite and high-resolution in-situ observations: satellites provide synoptic coverage, while local observations resolve small-scale processes that shape the isotopic structure of the atmosphere. Overall, our comparison with IASI should thus be regarded as a qualitative consistency check confirming large-scale and temporal coherence rather than a one-to-one correspondence.

  For most intervals, IASI satellite data closely match raw and corrected $\delta^2H$ measurements for altitudes 4000–7000m. In the 8000–12000m range, although applying the diffusion correction increased the correlation coefficient between the air-bag observations and the IASI data from 0.53 to 0.60, IASI data are lower than $\delta^2H$ air-bag observations during certain periods, particularly June and September 2020; one possible reason is unaccounted environmental influences at high altitudes. However, it is noteworthy that the IASI data closely match the observed $\delta^2H$ for all other periods in the 4000–7000 m range, they are also lower in June 2020. Moreover, diffusion within the sampling bags could, in principle, introduce systematic bias, yet the IASI values sometimes agree well, sometimes appear higher, and sometimes lower than our measurements (Fig. 10). Therefore, the differences between our in-situ observations and the IASI retrievals likely arise mainly from representativeness differences intrinsic to the satellite

data, as discussed above—practically the cloud-sampling limitation in the upper troposphere— rather than from diffusion-related biases. Accordingly, such variations cannot be corrected by the diffusion model, which accounts for isotopic fractionation during air-bag storage but not for spatiotemporal or cloud-sampling discrepancies between observation platforms.

The right panel of Fig.10 (Figs. 10 b, d, and f) shows the comparison of raw and corrected vapor d-excess measurements. No d-excess dataset is available for comparison for the 4000–7000m and 8000–12000m (Fig. 10d and f). As previously noted, corrected d-excess values should be and are lower than the raw data, as expected from diffusion theory. For the 8000–12000 m observations, the correction magnitude is relatively smaller than at lower altitudes due to the shorter storage time of the air bags.

Although no completely independent dataset is available for direct validation, our model— already verified by laboratory experiments—was used to perform a detailed assessment of all potential error sources and to quantify a comprehensive and conservative uncertainty range of the corrected data.

# 5  Conclusion

High spatial and temporal resolution water vapor isotope data are critical for understanding various hydrologic cycle processes. However, observations of vertical water vapor isotope profiles are scarce, particularly in the upper troposphere. Satellite-derived vapor isotope data are available only at limited vertical and temporal resolutions. Acquiring high-resolution water vapor isotope data, especially under conditions where direct measurements are difficult, has been a significant challenge for the water isotopes research community. This study demonstrates the potential of a drone-based air bag sampling method to overcome this challenge, offers solutions for evaluating air bag suitability and addressing air bag permeability.

While air bags offer the advantage of sample collection, their inherent permeability can affect the sealing integrity of the samples, leading to potential contamination. The permeability of airbag materials varies, with some exhibiting lower levels. We recommend prioritizing the use of glass containers and air bags with the lowest permeability for collecting water vapor using portable devices. Additionally, it is essential to conduct the permeability experiments described in this article before any experimental undertaking. This involves storing water vapor with known isotopic composition in the portable collection device for an extended period and then re-measuring these values to assess or determine the device's permeability parameters.

To further address the permeability challenge, we developed a mathematical model to evaluate and correct for diffusion and isotopic fractionation, ensuring the reliability of vapor isotope measurements using air bags. Calibrated with parameters from laboratory experiments, our correction model reconstructs the initial isotopic composition of sampled vapor by using data from both the air bag and the surrounding environment, offers a potential solution to the prevalent permeability challenges. This model was rigorously validated against observational experiments conducted under varying conditions. We also applied this model to drone-collected samples at various pressures. By estimating uncertainty and comparing corrected data with in-situ surface-level measurements on the Picarro and satellite observations, we validated the reliability and applicability of drone-based water vapor isotope measurements.

Our drone-based sampling system, combined with the diffusion model, effectively
addresses the limitations of traditional high-altitude water vapor measurement methods. It
meets the need for lightweight equipment while providing a more economical, efficient, and
flexible alternative to conventional approaches involving large aircraft, airships, and balloons.
This approach enables us to exploit the benefits of drone-based air bag sampling at high
altitudes and over long distances, while effectively mitigating its potential limitations. This
strategy significantly broadens its potential applications across various environments, thereby
enhancing the range and richness of data that can be gathered for water vapor isotope research.
This study was conducted under stable temperature and equal internal and external
pressures during storage. We acknowledge that a formulation based on partial pressure of water
vapor would be more general and could improve model applicability under varying temperature
and pressure conditions. Future work could extend the model to account for these factors.

## Appendix A


*Table A1 Summary of experiments: diffusion parameter quantification, model validation and differences in experimental methods. $\lambda_{surface}$ denotes the water vapor exchange coefficient at the surface, $\alpha$ refers to the fractionation coefficient of isotopes, $q_0$ represents the initial vapor humidity in the air bags, and $q_e$ corresponds to the environmental vapor humidity.*

| Experiment Number | Experimental purpose | Differences in experimental methods |
|---|---|---|
| No. 1 | Quantification of $\lambda_{surface}$ | Dry air in the air bags |
| No. 2 | Quantification of $\alpha$ | Water vapor with known isotopic compositions in the air bags under the condition $q_0 = q_e$ |
| No. 3 | Diffusion model validation | Water vapor with varying isotopic compositions and humidity levels in the air bags |


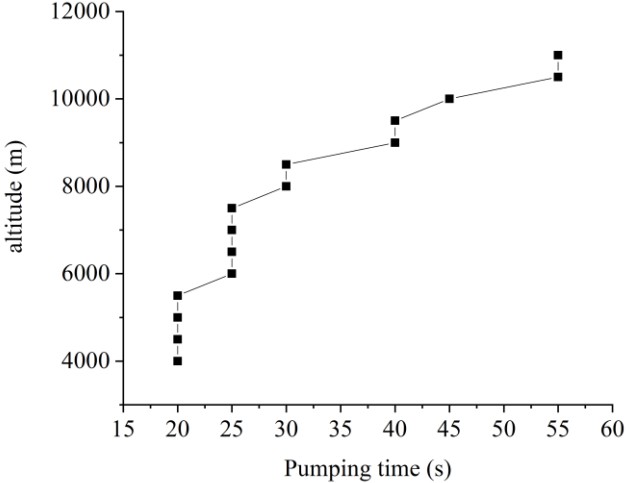


Figure A1 Sampling duration variation with altitude.

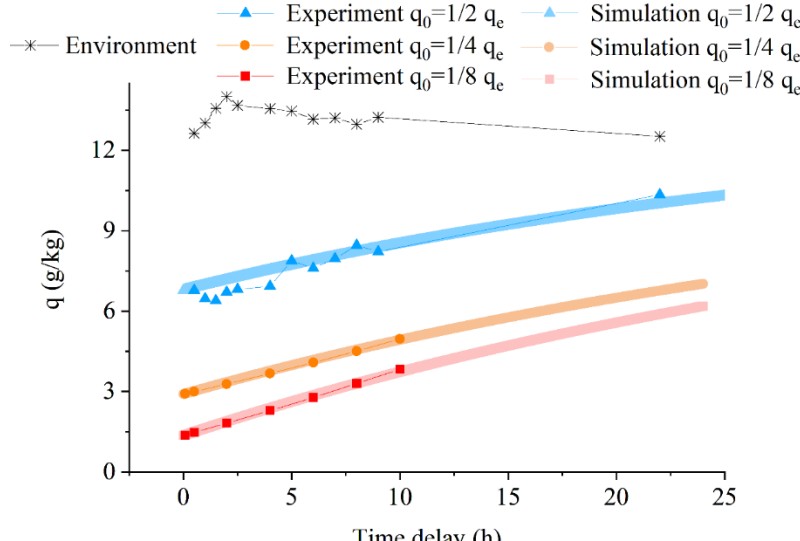


Figure A2 Comparison of humidity variation within the air bag over time in laboratory
permeability Experiments No.3 (markers) and diffusion model simulations (lines) for Figures
4 and 5. The experiments cover a range of initial humidities at $t = 0$ ($q_0$), from approximately

$1/8 * q_e$ to $1/2 q_e$, where $q_e$ denotes environmental humidity.


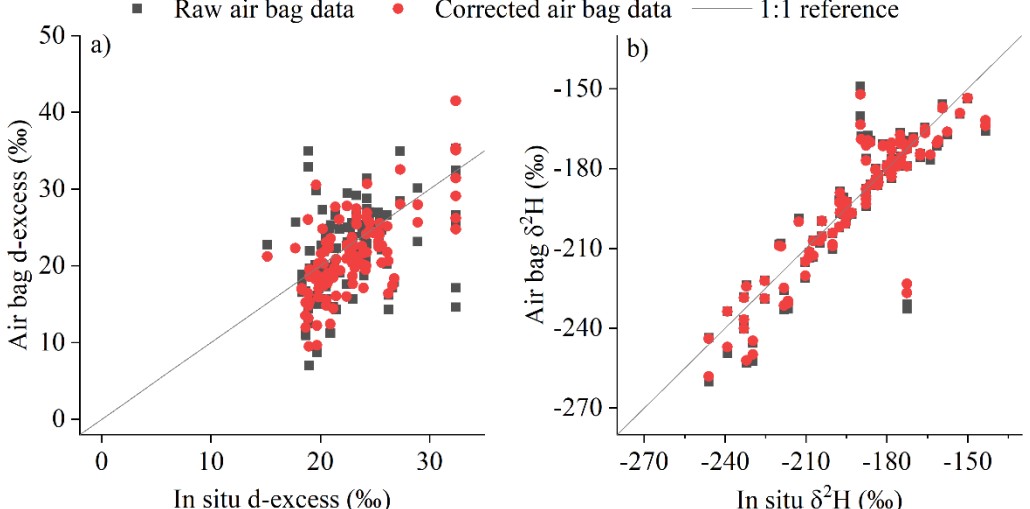


Figure A3 Comparison of raw and corrected air bag data with in situ observations averaged

over the sampling period (1:1 reference shown)

## Code availability

The diffusion modeling code, which simulates the evolution of water vapor isotopes in the
airbag and corrects them to their initial values, is available at:
https://github.com/DishiWANG0608/Correcting-for-water-vapor-diffusion.git

## Data availability

Data are available from the authors on request.

## Video supplement

A video showcasing our field campaign on drone-derived water vapor isotope sampling up to the upper troposphere (11 km) during convective activity, including the workflow for airbag water vapor isotope sampling, is currently available upon request and will be publicly accessible in the near future. Please contact the corresponding author for access (di.wang@latmos.ipsl.fr).

## Author contributions

D.W., C.R., and L.T. designed the research, established the subjects of the methodology, and performed the analysis; D.W. and D.Y. conducted the observations; G.B., S.F., H.P. and L.L. contributed to the establishment of methodologies, data calibration, and analysis; Y.S. provided assistance with the computing; All authors contributed to the discussion of the results and the final article; D.W. drafted the manuscript with contributions from all co-authors.

## Competing interests

The contact author has declared that none of the authors has any competing interests.

## Acknowledgments

The authors gratefully acknowledge the Infrared Atmospheric Sounding Interferometer (IASI) for providing spatiotemporal coverage of $\delta^2H$ retrieval. We thank Jiaping Xu, Mingjian Chen, Le Chan, and Yao Yao for their irreplaceable help in the development of the drones. We are grateful to the management of 99 Dragon, Mountain Laojun in Lijiang for facilitating the field experiments. We thank Pengbin Liang and Jiangrong Tai for partially participating in the field observations. We acknowledge Hans Christian Steen-Larsen and Harald Sodemann for their discussions on the correction method. We acknowledge the use of OpenAI's ChatGPT to assist in refining the language. This work was supported by the National Natural Science Foundation of China (Grants No. 42401043 and U2202208), the fellowship from the China Postdoctoral Science Foundation (Grant No.2025T180092), the Strategic Priority Research Program of the Chinese Academy of Sciences (Grant No. XDB40000000), and the Science and Technology Department of Yunnan Province (Grant No. 202201BF070001-021). Di Wang acknowledges support from Chinese Scholarship Council and the Postdoctoral Fellowship

Program of China Postdoctoral Science Foundation (Grant No. GZC20241440). Siteng Fan
acknowledges support from the Marie Skłodowska-Curie Actions of European Research
Executive Agency (Grant No. 101064814) and the High Level Special Fund of Southern
University of Science and Technology (Grant No. G03050K001).

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
