# Peer review of "Correcting for water vapor diffusion in air bag samples for isotope composition analysis: case study with drone-collected samples"

_Atmospheric Measurement Techniques, 2024_

## Author Comment (AC1)

**Response to Reviewers**

We sincerely appreciate the insightful comments and constructive suggestions from both reviewers, which have significantly improved the clarity and robustness of our methodology. We also thank the reviewers for their positive and encouraging feedback.

In this response letter, we have addressed each comment in detail below. Our responses are highlighted in blue, with corresponding revisions in the manuscript also indicated in *blue italics*. All line numbers in this document refer to the updated version of the manuscript.

Thank you for your time and consideration.

With best regards,
Di WANG
On behalf of all authors

**AC2: 'Comment on amt-2024-151', 16 Dec 2024**

Wang and others have developed a theoretical model to describe water vapour diffusion through the surface of a sampling bag. They calibrated the model's parameters using laboratory experiments. This model allows for the reconstruction of the initial isotopic composition of the sampled vapour by using measurements taken from both inside the bag and the surrounding environment. I believe it is an important work but not well explained and supported in several sections, and needs a through revision.

We appreciate your positive comment. We also appreciate your detailed suggestions.

L67 Distinguish between aerial mobile measurements for water vapour isotopes and mobile water isotope measurements, i.e. picarro mounted on a van. The latter is quite common and does not require sample storage anymore.

We have incorporated your suggestion by distinguishing aerial from mobile water isotope measurements (lines 64-67):

*"However, their heavy instrumentation, substantial power requirements, and limited mobility restrict their usability in certain situations, particularly for aerial water vapor isotope measurements, which require lightweight and flexible sampling approaches."*

L88-90 check usage; this should be isotopologues

We have replaced 'isotopes' with 'isotopologues'.

L104 What diverse conditions?

We have replaced 'diverse conditions' with 'varying humidity and isotopic

composition differences between the inside and outside of the air bag' (line 106)

L110-111 Picarro direct observations - as in measurement on the Picarro? And satellite data of what?

To provide a more precise description of the data used, we have revised lines 112–117 as follows:

*"The corrected near-surface drone-based measurements using our diffusion model show consistency with direct, in-situ surface-level measurements using the Picarro analyzer. Similarly, at two mid-tropospheric levels, the corrected drone measurements align with IASI satellite observations of water vapor isotopic composition, further confirming the model's theoretical and practical reliability in applications."*

L125 State the boundary conditions and assumptions clearly under which equation 1 is valid

We appreciate your advice and have added the boundary conditions and assumptions for Equation 1 and 2 (lines 139-142):

*"The validity of Equation (1) and (2) relies on the assumptions that internal and external pressures remain equal to atmospheric pressure, ensuring no pressure gradient across the bag membrane, that the internal vapor is well-mixed, and that the exchange rate follows a first-order process. Additionally, if the temperature remains constant, $k$ and $k_i$ are assumed to be constant."*

L130 Equation 2 should be based on isotope notations, and the isotopologue for which they apply can be in subscript. This is not clear yet.

We appreciate the reviewer's advice. While δ-values provide a clearer visualization of isotope variations, their direct use in equations obscures the model's physical basis and complicates interpretation of fractionation processes. To maintain mathematical consistency and align with standard fractionation factor definitions, we use isotope ratios (R) instead of δ-values.

To ensure the correct use of R and δ-values in different contexts, we have now explicitly stated how R and δ-values are used in the manuscript (lines 212-216):

*" For mathematical clarity and consistency, isotopic ratios (R) are used in the equations presented in previous sections. Replacing R with δ-values would only shift the physical basis without affecting the mathematical validity of the equations or the estimation of α, as the standard ratio cancels out. For clearer visualization, δ-values are used for numerical applications and in the subsequent figures and tables."*

L202 Convert the flow rate to volume per unit time

We have added the following clarification to explicitly state the flow rate (230-233):

*"In the measurement procedure, we first activated the dry air cylinder and adjusted the pressure reducing valve to 2 psi (pounds of force per square inch), within the Picarro water isotope analyzer's recommended range of 2–4 psi for carrier gas. The instrument's built-in flow regulation maintains a gas flow rate of 30–50 mL/min,*

*ensuring stable sample delivery."*

L228 Provide more details on the experimental setup, including the type of airbags used and the specific model of the Picarro analyzer

We have provided the type of airbags used and the specific model of the Picarro analyzer (lines 219-221):

*"In this study, we used 0.5 L and 4L Teflon air bags produced by Dalian Hede Technologies Co., Ltd to collect and store vapor, and measured the vapor isotopes using a Picarro 2130i water isotope analyzer."*

L252 Provide more details on the experimental setup, such as the specific amounts of water injected and the isotopic values used, to give a clearer picture of the conditions tested.

We have revised the text to include the specific amounts of water injected and the isotopic values used (lines 294-301):

*"To validate the diffusion model under diverse conditions and evaluate its uncertainties, we repeated Experiment No. 2, but injected different amounts of water with known isotopic values to achieve a range of humidities from approximately 1/8 \* $q_e$ to $q_e$. Using the method described in Experiment No. 2, we injected 6 to 50 µL of reference water into a 4L air bag filled with dry air to achieve the desired humidity range. Additionally, we repeated the experiment using two reference waters with distinct isotopic compositions, specifically $\delta^{18}O$ = -58.07‰, $\delta^2H$ = -447.41‰ and $\delta^{18}O$ = -29.84‰, $\delta^2H$ = -222.84‰. To assess extended-duration variations, we also lengthened the time interval to 24 hours."*

L277 Drone flight path and sampling strategy need to be better explained. Also, samples aren't measured in situ.

We appreciate the reviewer's suggestion and have provided more details about the drone flight path and sampling strategy (lines 328-337):

*"We collected water vapor samples every 500 meters, starting from near the surface along the vertical profile. To optimize sampling across different altitude ranges, we deployed UAVs designed for varying flight altitudes. Generally, the UAV operating at lower altitudes collected samples at seven heights from 4,000 to 7,000 meters in a single flight. The mid-altitude UAV collected samples at four heights from 7,500 to 9,000 meters in one flight, while the high-altitude UAV collected samples at four heights from 9,500 to 11,000 meters in two flights. Each flight took approximately 20~30 minutes. In case of any disruptions during sampling, we repeated the process until a complete vertical profile was obtained. At the beginning of the experiment, we also collected replicate samples at each height to ensure data consistency."*

To clarify our analysis procedure, we revised Line 338-340 as follows:

*"By integrating high-altitude drone sampling with subsequent water vapor isotope analysis using the Picarro analyzer at the surface, we obtained vapor isotopic profiles up to an altitude of 11 km."*

L305 I would suggest a net uncertainty or error propagation of some kind to be calculated and reported for

λ_surface, α_δ, λ_alt. Currently, this section does not explain the uncertainties (only how they are calculated) or how they affect the results.

We have revised the manuscript to explicitly describe how uncertainties from all sources were combined to calculate the net uncertainty. First, we clarified that as the pressure and sampling time are well known, the uncertainty in λ_alt is fully propagated from the uncertainties in λ_surface and pump efficiency ($\varepsilon$), rather than being treated as an independent source, as shown in Equation 15 (Section 3.4.1):

$$\lambda\_alt = \lambda\_surface * \frac{P\_surface}{P\_alt} * \frac{Sampling\ time\_alt}{Sampling\ time\_surface} * \varepsilon$$

We have ensured that Section 3.4.2 ("The Method of Uncertainty Estimation") details the calculation methods for λ_surface, α, and pump efficiency ($\varepsilon$), while Section 4 ("Discussion") explicitly discusses how these uncertainties influence the results.

We also expanded the discussion in Sections 4.3 and 4.5 to explicitly report the net uncertainties for δ18O and d-excess across all altitudes (lines 569-570):

*"The combined uncertainty from all sources, including λ_surface, α, pump efficiency (ε), and model-experiment mismatches, results in a total uncertainty of approximately 1‰ for δ18O and 8‰ for d-excess across 98% of the data."*

Fig 4 How are the equilibriation lines intersecting? And mention which experiment generates this.

We now provide an interpretation of this result. To achieve this, we slightly modified the final form of Eq.8 to isolate two terms: the first term drives $R_{(t)}$ towards $R_e$ at a constant rate, while the second term drives $R_{(t)}$ towards $R_e/\alpha$ at a rate dependent on $(q_e-q)$ (line 206).

$$\frac{dR(t)}{dt} = \frac{\lambda}{\alpha} * \left( R_e - R(t) \right) + \frac{\lambda}{q(t)} * \left( q_e - q(t) \right) * \left( \frac{R_e}{\alpha} - R(t) \right) \qquad (8)$$

We have added the following discussion in the manuscript (lines 459-466):

*"We observe that some curves unexpectedly intersect, which can be understood by analyzing Eq. (8). The first term ($\frac{\lambda}{\alpha}(R_e-R_{(t)})$) continuously drives $R_{(t)}$ towards $R_e$, while the second term ( $\frac{\lambda}{q(t)}(q_e-q_{(t)})(\frac{Re}{\alpha}-R_{(t)})$) modulates the rate of change. Initially, for simulations with lower $q_0$ values (e.g., the red and orange curves), $q_e-q_{(t)}$ is large, making the second term significant and positive, thereby increasing $R_{(t)}$ more rapidly. However, as $R_{(t)}$ exceeds $R_e/\alpha$, the sign of this term reverses, slowing down the increase in $R_{(t)}$ compared to other curves. In contrast, curves with higher initial $q_0$ values (e.g., blue curve) experience a steadier growth and eventually surpass the initially faster-growing curves, leading to the observed crossing."*

We have specified which experiment generated the data (line 448):
*"To validate the model, we used Experiment No.3 described in Subsection 3.2."*

L441 explain why <1 permil is unrealistic

We have removed the following sentences from the manuscript and retained data points with d-excess values less than 1‰:

"In this dataset, acquired from the drone observations and subsequently corrected using the diffusion modeling, data points with d-excess values less than 1‰ were omitted, as these values are unrealistic and likely result from overcorrection of the δ-values. This resulted in the exclusion of 6 out of 1039 samples."

This revision only affects a very small number of data points (6 out of 1039), and including these data does not alter the overall results and conclusions.

L460 I would explain this in the methods and bring it back in the discussion as a model sensitivity to its parameters

We appreciate the reviewer's advice. The method for error calculation is detailed in Section 3.4.2 ("The Method of Uncertainty Estimation"), while Sections 4.3 to 4.4 focus on interpreting the results. The full range of uncertainties listed in Table 1 is incorporated into the analysis, inherently capturing the model's sensitivity to its parameters. The final uncertainties in Subsections 4.3 to 4.5 reflect this combined maximum error.

To clarify this, we have revised the manuscript as follows (lines 388-391):

*"The maximum discrepancy across all calibration results—using the full uncertainty range for $\lambda_{surface}$, $\alpha$, and pump efficiency ($\varepsilon$)—was determined. The model-experiment mismatch was then added as an independent error component. The final uncertainty estimates, reported in Subsections 4.3 to 4.5, account for all potential error."*

Additionally, we have expanded the discussion to explicitly highlight the model's sensitivity to its parameters (lines 570-575):

*"Among these sources, $\varepsilon$ contributes the largest uncertainty, particularly at higher altitudes (Figure 8 and 9), likely due to the conservative uncertainty range we applied to account for potential reductions in collected air mass at high altitudes. Additionally, the fractionation coefficient ($\alpha$) also contributes considerably to the total uncertainty. To mitigate this, we recommend conducting multiple measurements to obtain an averaged value and performing repeated parameter validation to ensure robustness."*

L485 How different are these storage times to really affect the measurements? Can this be incorporated as part of the correction in the model?

The storage duration of air bags typically ranges from 10 minutes to 2 hours. The actual storage time was recorded for each sample and incorporated as a variable parameter in the model. This ensures that the effect of varying storage times on the final measurements is explicitly accounted for in the correction process.

To clarify, we added the following explanation and equation (lines 198-211):

*"The constants ($\lambda$, $\alpha\_{^{18}O}$, $\alpha\_{^2H}$) can be determined through laboratory experiments and Equations 10 and 13 (see Subsection 3.2 and 4.1). If we know the initial values within the air bag ($q_0$, $R\_{^{18}O_0}$, $R\_{^2H_0}$), the ambient values ($q_e$, $R\_{^{18}O_e}$, $R\_{^2H_e}$), and the storage time ($T_{storage}$) of the sampling bag, we are able to simulate the*

*variations in humidity and isotopic ratios inside the air bag according to Eqs. 5 and 8. Similarly, if we know T_storage, the humidity and isotopic values at time t = T_storage (q(T_storage), R_$^{18}$O(T_storage), R_$^{2}$H (T_storage) in the air bag, and the ambient values, we can deduce the initial values in the air bag at t = 0 by back-calculating. The equation used for reconstructing the initial isotope ratio (R$_0$) is:*

$$R_0 = R_{measured} - \int_0^{T_{storage}} \frac{dR(t)}{dt} dt$$

$$= R_{measured} - \int_0^{T_{storage}} (\frac{\lambda}{\alpha} * (R_e - R(t)) + \frac{\lambda}{q(t)} * (q_e - q(t)) * (\frac{R_e}{\alpha} - R(t))) dt \quad (14)$$

*where R$_0$ represents the initial isotopic ratio we want to reconstruct, R_measured is the observed isotopic ratio after T_storage, and $\frac{dR(t)}{dt}$ is defined in Eq.8.*

*This approach allows us to correct for diffusion-induced isotopic shifts and reconstruct the original vapor composition."*

L490 This section has been introduced several times in the paper but is not discussed enough here. I would expect some prior information about why they may be different based on the remote sensing method but necessary to fit wider regions or global models. I would also expect the authors to mention other such repositories like TES and SCIAMACHY.

We appreciate the reviewer's suggestion and have revised the manuscript to provide additional context on the differences between satellite-derived and in-situ measurements. We now clarify that (lines 408-420):

*"Satellite measurements, particularly for vertical profiles of water vapor isotopes, are inherently different from direct sampling, they represent a vertical average over layers determined by the averaging kernels (Rodgers and Connor, 2003; Worden et al., 2006). Therefore, their comparability with ground-based or drone-based observations, which provide high-resolution local data, is limited. In this study, we use the MUSICA retrievals from the IASI satellite instrument (Diekmann et al 2021), which provides water vapor isotope data at three altitude levels: 1-3 km in the lower troposphere, 4-7 km in the mid-troposphere, and 8-12 km in the upper troposphere. Given that our study started at an altitude of 3856 m, we used the retrieved $\delta^2 H$ data for the 4–7 km and 8–12 km levels. However, these measurements represent a vertical average over layers determined by the averaging kernels (Rodgers and Connor, 2003; Worden et al., 2006). While using averaging kernels to smooth the observed profile could facilitate a more quantitative analysis, we simply averaged the observations for the corresponding altitudes. Consequently, the comparison remains mainly qualitative."*

Additionally, we have expanded the discussion to acknowledge other satellite retrievals, including TES and SCIAMACHY, as follows (lines 394-402):

*"Several satellite missions have contributed to water vapor isotope observations,*

*including the Tropospheric Emission Spectrometer (TES) onboard Aura (2004–2018) (Worden et al., 2006), the Scanning Imaging Absorption Spectrometer for Atmospheric Cartography (SCIAMACHY) onboard Envisat (2002–2012), the Atmospheric Infrared Sounder (AIRS) onboard Aqua (since 2002) (Worden et al., 2019), and the Tropospheric Monitoring Instrument (TROPOMI) onboard Sentinel 5 Precursor (since 2017) (Schneider et al., 2022). In this study, we use the MUSICA retrievals from the Infrared Atmospheric Sounding Interferometer (IASI) onboard METOP due to its broad spatiotemporal coverage, vertical profiling capability, and the availability and accessibility of its dataset (Diekmann et al., 2021)."*

Fig 10 The left panels are of d2H, but the figure caption and subsequent discussion on d18O. I expect the satellite data to be that of d2H. What am I missing here?

We exclusively compared and discussed $\delta^2H$ in this section, as satellite data is only available for $\delta^2H$. I have revised the discussion section accordingly. Thank you for the correction.

Fig 10e Explain why, for higher elevation samples, the satellite dD differs more with measured/corrected data than other altitudes.

We have added an explanation in the manuscript to clarify this discrepancy (lines 543-551):

*"Errors due to uncertainties in pump efficiency ($\varepsilon$) are the main source, exhibiting the largest spread (Fig. 8) and increasing with altitude (Fig. 9). Errors derived from $\lambda_{\_surface}$ and $\alpha$ also increase with altitude (Figs. 8 and 9). As a result, at higher elevations, the satellite $\delta^2H$ differs more from the measured and corrected data than at lower altitudes (Fig. 10). This pattern arises because $\lambda_{\_alt}$ deviates more from $\lambda_{\_surface}$ at higher elevations (Eq.16), primarily due to increased errors in estimating $M_{\_alt}$, amplifying correction errors. Moreover, the humidity and isotopic disparity between the air captured in the air bag and lower-altitude ambient air widens with altitude, requiring more intensive corrections. Consequently, both the uncertainty (Figs. 8 and 9) and the magnitude of the diffusion correction (Fig. 7) increase with altitude."*

---

## Author Comment (AC2)

**Response to Reviewers**

We sincerely appreciate the insightful comments and constructive suggestions from both reviewers, which have significantly improved the clarity and robustness of our methodology. We also thank the reviewers for their positive and encouraging feedback.

In this response letter, we have addressed each comment in detail below. Our responses are highlighted in blue, with corresponding revisions in the manuscript also indicated in *blue italics*. All line numbers in this document refer to the updated version of the manuscript.

Thank you for your time and consideration.

With best regards,
Di WANG
On behalf of all authors

**AC1: 'Comment on amt-2024-151', 16 Dec 2024**

This manuscript describes methods by which to interpret isotope and humidity data from drone-mounted permeable gas sampling bags. The conceptual model appears to be generally useful for inferring atmospheric conditions based on gas composition inside sampling bags. However, the conceptual derivation of the model should be more robustly described, improving precision of communication but especially considering all relevant effects such as temperature and pressure. There is also a lot of improvement needed in precision of notation and terminology, such as the definition of diffusion, silent substitution of delta notation for isotopic ratio in a key equation, and others that I have noted below in the detailed comments. Finally, the actual sampling techniques are not described, so the example drone flight profile is difficult to interpret.

We appreciate the reviewer's thoughtful evaluation of our study and the detailed suggestions provided. Your feedback has helped us refine the conceptual model, improve the clarity of our manuscript. We have strengthened the conceptual derivation of the model, improved the notation and terminology for consistency, and provided a more detailed description of the sampling techniques. Below, we address each comment in detail.

L90 differential diffusion does cause fractionation, but they are not synonymous.

To avoid ambiguity, we have revised the sentence as follows (lines 91-92):

*"This differential diffusion, can alter the original isotopic composition of the collected air samples."*

L90 there are two relevant gradients causing fractionation: one in concentration of water (causing mass flux), and one in isotopic composition (resulting in no net mass flux). The latter can cause fractionation even if humidity is the same inside and outside of the bag. Neglecting the second gradient may be justified if it is small, but it should

not be ignored completely by the theoretical derivation.

We acknowledge that the role of isotopic composition gradients was not explicitly stated in the Introduction. To address this, we have added the following clarification (lines 92-93):

*"Moreover, differential diffusion can also occur due to gradients in isotopic composition."*

Additionally, we would like to point out that our model (Eq. 8) accounts for both the concentration gradient ($q_e$ - q(t)) and the isotopic composition gradient ($R_e$ - R(t)).

L125 I have several comments about Eq 1:

it neglects effects of pressure and temperature differences across the bag membrane. (2) wouldn't it be more general to formulate this equation in terms of partial pressure of water vapor instead of mass concentrations? That would partially resolve #1

We appreciate the reviewer's insightful comments.

Regarding pressure and temperature differences across the bag membrane:

The air bags were stored and measured in a temperature-controlled chamber, ensuring stable temperature conditions throughout the experiment (lines 226-228). Internal and external pressures were equal to atmospheric pressure, meaning no pressure gradient existed across the bag membrane. To clarify this, we have explicitly stated the boundary conditions and assumptions under which Equations (1) and (2) hold (lines 139-142):

*"The validity of Equation (1) and (2) relies on the assumptions that internal and external pressures remain equal to atmospheric pressure, ensuring no pressure gradient across the bag membrane, that the internal vapor is well-mixed, and that the exchange rate follows a first-order process. Additionally, if the temperature remains constant, k and  $k_i$ are assumed to be constant."*

Regarding the use of partial pressure instead of mass concentration:

This is indeed a valuable approach that could enhance the general applicability of the model, particularly under conditions where pressure and temperature gradients exist across the bag membrane. Under our experimental setup, where temperature is stable and internal and external pressures are equal, mass concentration and partial pressure are directly proportional. As a result, using mass concentration is fully consistent with our experimental conditions. This formulation simplifies the parameterization and is more convenient for comparison with direct humidity measurements. We fully agree that future work could extend the model to incorporate varying pressure and temperature conditions, where a formulation based on partial pressure would be more appropriate. We have noted this as a potential future direction in the revised manuscript (lines 635-638):

*"This study was conducted under stable temperature and equal internal and external pressures during storage. We acknowledge that a formulation based on partial pressure of water vapor would be more general and could improve model applicability under varying temperature and pressure conditions. Future work could extend the*

*model to account for these factors."*

L124 It's not flux toward the bag, but into the bag, correct?

The phrase "The flux of water toward the bag " has been modified to "The flux of water into the bag" to correctly describe the direction of water movement (line 127).

defining k in g/kg adds a potentally confusing dimensionality to k, so it is not dimensionless as might be assumed and so that F takes on the units of g/m2/s instead of the si base units kg/m2/s that readers might assume without reading carefully. Maybe there is a good reason for the choice; please tell us.

We appreciate your comment. To ensure consistency with SI units, we have adjusted the units of the parameters in this equation. Specifically, we changed the units of $q_{(t)}$ (the variation of humidity inside the air bag over time) and $q_e$ (the environmental humidity) from g/kg to kg/kg (lines 127-131).

To reflect this change, we revised the equation as follows:

*"The flux of water into the bag, F (in kg/m²/s), is expressed as:*

$$F = k * (q_e - q(t)) \hspace{4cm} (1)$$

*where q(t) represents the variation of humidity inside the air bag over time (in kg/kg), $q_e$ denotes the environmental humidity (in kg/kg), k is water vapor conductance."*

L133 and L141 the definitions of alpha and lambda are both crucial equations. Assigning them equation numbers would make them easier to find.

We have now assigned equation numbers to the definitions of α and λ in the revised manuscript to improve clarity and accessibility (lines 138 and 154).

L139   the mass balance assumptions are not clear. How can M be constant if there is water flux into the bag? Does this mean we must assume that all vapor transport into the bag is balanced by an equal mass of non-water vapor transport out of the bag? Or maybe this theory only works if dM/dq is very small? What are the limits of this assumption?

We sincerely appreciate the reviewer's insightful question regarding the mass balance assumption. The assumption that M remains constant is based on the fact that specific humidity (q, in kg/kg) is much smaller than 1, meaning water vapor contributes only a small fraction of the total air mass. Even under extreme conditions, q varies from approximately $0.5 \times 10^{-3}$ (0.1%) to $13 \times 10^{-3}$ (1.3%) along the vertical profile, leading to a maximum mass variation of ~1%. This variation is negligible compared to the total air mass, making the assumption of constant M a reasonable approximation.

To improve clarity, we have revised the manuscript as follows (lines 149-150):

*"Assuming that M is constant, which is reasonable given that the total mass variation due to water vapor flux is at most 1%."*

L141 this definition of lambda can be loosely defined as a diffusion coefficient, but it is probably better termed (non-dimensional) conductance. It defines the rate of net mass flux in response to a gradient in concentration, but neglects gradients in pressure,

temperature, and isotopic composition. A standard definition of a diffusion coefficient would have dimensions cm2/s. Further confusion arises in sec 3.2, where lambda is called "permeability" L218. Please choose consistent terms.

We appreciate the reviewer's clarification regarding terminology. To improve consistency and precision, we have standardized the nomenclature throughout the manuscript as follows:

1) k is now referred to as water vapor conductance.
2) ki is now referred to as isotopic conductance.
3) λ (= k*A/M) is now consistently referred to as the water vapor exchange coefficient, incorporating both the exchange area and the air mass inside the bag.
4) λ/α is now consistently referred to as the isotopic exchange coefficient.

L167 it would be helpful to be extra clear here that the alphas being obtained are those due to fractionation due to mass flow through bags.

We accepted your suggestion and have revised the sentence as follows (lines 182-183):

*"Knowing λ, we can deduce the isotopic fractionation coefficient due to fractionation caused by mass flow through bags, α, for each isotope."*

L181-183 there is a lack of clarity here in notation. Delta 18O and delta 2H do not appear in eq 11. Substituting delta notation for ratio notation has no consequences for the alpha in Eq 11 because the standard ratio cancels out, but it would be kinder to readers to justify the use of delta notation either by deriving Eq 11 in terms of deltas or to explain here that ratios of R and ratios of deltas are equivalent.

We appreciate the reviewer's suggestion for improving clarity in the notation. To maintain mathematical consistency and align with standard fractionation factor definitions, we use isotope ratios (R) instead of δ-values in equations. To improve accessibility for readers, we have modified and included the following explanation in the manuscript (lines 198-216):

*"The constants (λ, α_$^{18}$O, α_$^2$H) can be determined through laboratory experiments and Eqs. 10 and 13 (see Subsection 3.2 and 4.1). If we know the initial values within the air bag ($q_0$, R_$^{18}$O$_0$, R_$^2$H$_0$), the ambient values ($q_e$, R_$^{18}$O$_e$, R_$^2$H$_e$), and the storage time ($T_{storage}$) of the sampling bag, we are able to simulate the variations in humidity and isotopic ratios inside the air bag according to Eqs. 5 and 8. Similarly, if we know $T_{storage}$, the humidity and isotopic values at time t = $T_{storage}$ ($q_{(T\_storage)}$, R_$^{18}$O$_{(T\_storage)}$, R_$^2$H$_{(T\_storage)}$ in the air bag, and the ambient values, we can deduce the initial values in the air bag at t = 0 by back-calculating. The equation used for reconstructing the initial isotope ratio ($R_0$) is:*

$$R_0 = R_{measured} - \int_0^{T_{storage}} \frac{dR(t)}{dt} dt$$

$$= R_{measured} - \int_0^{T_{storage}} \left( \frac{\lambda}{\alpha} * (R_e - R(t)) + \frac{\lambda}{q(t)} * (q_e - q(t)) * \left( \frac{R_e}{\alpha} - R(t) \right) \right) dt$$

*(14)*

*where $R_0$ represents the initial isotopic ratio we want to reconstruct, $R_{measured}$ is the observed isotopic ratio after $T_{storage}$, and $\frac{dR(t)}{dt}$ is defined in Equation (8).*

*This approach allows us to correct for diffusion-induced isotopic shifts and reconstruct the original vapor composition.*

*For mathematical clarity and consistency, isotopic ratios (R) are used in the equations presented in previous sections. Replacing R with δ-values would only shift the physical basis without affecting the mathematical validity of the equations or the estimation of α, as the standard ratio cancels out. For clearer visualization, δ-values are used for numerical applications and in the subsequent figures and tables."*

L202 flow rate is not measured in psi.

We have added the following clarification (lines 230-233):

*"In the measurement procedure, we first activated the dry air cylinder and adjusted the pressure reducing valve to 2 psi (pounds of force per square inch), within the Picarro water isotope analyzer's recommended range of 2–4 psi for carrier gas. Due to built-in flow regulation, the instrument maintains a gas flow rate of 30–50 mL/min."*

L209 "can" or "did"? And what is a parallel sample?

We have revised "can achieve" to "achieved" for accuracy and replaced "parallel samples" with " replicate samples—air samples collected simultaneously under the same conditions" for clarity.

L210 I'm not following—which bias is this? I am guessing this is in the laser spec, but it would be nice to be clear.

We have clarified that the correction addresses isotope measurement bias due to the instrument's sensitivity to water vapor concentration (lines 242-243):

*"To correct the isotope measurement bias caused by the instrument's sensitivity to different water vapor concentrations (Schmidt et al., 2010)"*

L210 this paragraph is difficult to follow because it appears to use jargon specific to the piece of equipment used (but not fully specified—was it a Picarro A0101?).

We have revised the paragraph for clarity and explicitly provided the specific model of the Picarro analyzer. The updated text now states (lines 242-247):

*"To correct isotope measurement bias caused by the instrument's sensitivity to different water vapor concentrations (Schmidt et al., 2010), we used the built-in Standard Delivery Module (SDM) of the Picarro 2130i water vapor isotope analyzer to generate a 500–25,000 ppm water vapor gradient for isotope measurements. We selected 20,000 ppm as a reference humidity level, as this corresponds to the optimal accuracy range of the Picarro analyzer (JingfengLiu et al., 2014; Schmidt et al., 2010)."*

Table 2.1 is not needed

We have moved the table to Appendix A.

L228 alpha_delta is an unfortunate choice in nomenclature. The standard variable is alpha, which can be made more specific by listing the isotopes involved (eg alpha H2/H1), but it adds only confusion to add "_delta" because the delta notation has nothing to do with the isotope fractionation factor. Alpha is defined in terms of isotopic ratios (i.e., not in terms of delta).

To avoid confusion, we have revised the notation as follows:

$\alpha\_\delta \rightarrow \alpha$

$\alpha\_\delta^{18}O \rightarrow \alpha\_^{18}O$

$\alpha\_\delta^2H \rightarrow \alpha\_^2H$

L232 injected how? Liquid? This is ~$10^{-2}$ ml, correct? L255 And $10^{-3}$ ml in experiment 3? I'm surprised this was easier than using lab air and later adjusting the humidity or isotopic composition of a testing chamber.

We have supplemented the manuscript with a detailed explanation of how reference air bags with known water vapor isotopic values were prepared. The revised text now states (lines 267-274):

*"Empty, clean air bags were first filled with dry air, then sealed by closing the bag valve. To maintain a closed system while injecting reference water, a dedicated injection septum was installed on the valve. After reopening the valve, a fixed amount of laboratory reference liquid water with known isotopic values was injected into the dry air-filled bag using a 10 μL injection needle. In Experiment No. 2, we ensured that the initial humidity ($q_0$) was approximately equal to the environmental humidity ($q_e$). To ensure $q_0 = q_e$, the environmental vapor concentration was first measured, followed by the calculation and experimental determination of the water volume to be injected into the air bag."*

We also have revised the text to include the specific amounts of water injected and the isotopic values used (lines 293-299):

*"To validate the diffusion model under diverse conditions and evaluate its uncertainties, we repeated Experiment No. 2, but injected different amounts of water with known isotopic values to achieve a range of humidities from approximately 1/8 \* $q_e$ to $q_e$. Using the method described in Experiment No. 2, we injected 6 to 50 μL of reference water into a 4L air bag filled with dry air to achieve the desired humidity range. Additionally, we repeated the experiment using two reference waters with distinct isotopic compositions, specifically $\delta^{18}O$ = -58.82‰, $\delta^2H$ = -428.82‰ and $\delta^{18}O$ = -29.89‰, $\delta^2H$ = -222.89‰."*

L259 "can" or "did"?

We have revised "can used" to "used" for accuracy.

L266-268 the logic of how the sampling system works is important. Full details might not be appropriate here, but a citation to them would be nice. At minimum I would expect an outline of how it works, given that understanding the results depends on

understanding the methods.

We appreciate your suggestion and have supplemented the paragraph with a more detailed description of the sampling system (lines 311-317):

*"We designed and built a collection module for fixed-height sampling, incorporating diaphragm vacuum pumps, a rudder mounted on the drone, and a control module linked to a remote operating system. When the drone reaches a specified altitude, we remotely activate the designated air pump to inflate a specific air bag. Once sampling is complete, the pump is deactivated, and the drone ascends to the next target altitude, where the corresponding air pump inflates another air bag. This process was repeated until all predetermined samples were collected."*

We also have provided more details about the drone flight path and sampling strategy in the following paragraph (lines 328-337):

*"We collected water vapor samples every 500 meters, starting from near the surface along the vertical profile. To optimize sampling across different altitude ranges, we deployed UAVs designed for varying flight altitudes. Generally, the UAV operating at lower altitudes collected samples at seven heights from 4,000 to 7,000 meters in a single flight. The mid-altitude UAV collected samples at four heights from 7,500 to 9,000 meters in one flight, while the high-altitude UAV collected samples at four heights from 9,500 to 11,000 meters in two flights. Each flight took approximately 20~30 minutes. In case of any disruptions during sampling, we repeated the process until a complete vertical profile was obtained. At the beginning of the experiment, we also collected replicate samples at each height to ensure data consistency."*

L269 of course the bags do not deflate because of mass loss, but because of increased pressure outside the bag, and the pressing danger would therefore seem to be preventing ingress of new air, not egress of sample. Do the one-way valves protect against this?

We appreciate the reviewer for pointing out our oversight in our writing and raising the concern about air ingress. To address this, we have added details about the vacuum diaphragm pump (lines 317-323):

*"Self-sealing diaphragm vacuum pumps were used to transfer air into the sampling bags. Once the pump ceased operation, it remained sealed from the external environment, preventing unintended air ingress. Additionally, due to the flexible nature of the air bags, internal and external pressures remained balanced. As air pressure increases during the drone's descent after collection. To further prevent the loss of collected air samples, a one-way valve was installed to block backflow. Additionally, the one-way valve helps prevent large droplets from entering the air bag during the collection process."*

L280 the vapor is not measured in situ. Samples are removed from their locations and measured elsewhere.

We appreciate the reviewer's clarification. To clarify our analysis procedure, we revised Lines 337-339 as follows:

*"By integrating high-altitude drone sampling with subsequent water vapor isotope*

*analysis using the Picarro analyzer at the surface, we obtained vapor isotopic profiles up to an altitude of 11 km."*

L287 this sentence illustrates what I mean in my comment L266: readers cannot appreciate the sampling environment in any useful detail, so there is no way to fully understand why, for example, "it is difficult to experimentally estimate λ for different altitudes".

We have supplemented the paragraph with a more detailed description of the sampling system and strategy in response to Comment L266.

Here, we further clarify that we estimate M at high altitudes to determine $\lambda_{\_alt}$, while avoiding potentially confusing expressions (lines 345-350):

*"λ is defined as k*A/M and depends on the air mass M in the bag. In drone-based vertical sampling, M varies with altitude due to pressure changes, requiring an estimate of λ for different altitudes ($\lambda_{\_alt}$). However, since λ is an intrinsic property of the bag material, its apparent variation reflects uncertainties in estimating collected M, which depend on atmospheric pressure (P), sampling time, and pump efficiency (ε):*

$$M_{\_alt} = M_{\_surface} * \frac{P_{\_surface}}{P_{\_alt}} * \frac{Sampling\ time_{\_alt}}{Sampling\ time_{\_surface}} * \varepsilon \qquad (15)"$$

L291 longer sampling time where? To collect the air or to analyze the samples?

To specify that the extended sampling time applies to air collection at higher altitudes, we have revised the sentence as follows (lines 353-355):

*"To compensate for this effect, a longer sampling time was used to collect air at higher altitudes (Sampling time $_{\_alt}$) than at the surface (Sampling time $_{\_surface}$) (Fig. A1)."*

In general, section 3.4.1 seems to all collapse to "mass was estimated proportional to pressure at the sampling altitude and pumping time". The overly detailed presentation makes the logic seem more complicated than it is.

We have simplified Section 3.4.1 to improve clarity and avoid unnecessary complexity (lines 344-363):

*"3.4.1 Estimating the air mass in the bag*

λ is defined as k*A/M and depends on the air mass M in the bag. In drone-based vertical sampling, M varies with altitude due to pressure changes, requiring an estimate of λ for different altitudes ($\lambda_{\_alt}$). However, since λ is an intrinsic property of the bag material, its apparent variation reflects uncertainties in estimating collected M, which depend on atmospheric pressure (P), sampling time, and pump efficiency (ε):

$$M_{\_alt} = M_{\_surface} * \frac{P_{\_surface}}{P_{\_alt}} * \frac{Sampling\ time_{\_alt}}{Sampling\ time_{\_surface}} * \varepsilon \qquad (15)$$

where $M_{\_alt}$ is the air mass collected at a different altitude and $M_{\_surface}$ represents the air mass collected at the surface. At higher altitudes, where the air pressure ($P_{\_alt)}$ is lower than at the surface ($P_{\_surface}$), less air will be pumped into the air bag. To compensate for this effect, a longer sampling time was used to collect air at higher altitudes (Sampling time $_{\_alt}$) than at the surface (Sampling time $_{\_surface}$) (Fig.A1).

Given that $\lambda_{\_alt}$ is proportional to $M_{\_alt}$, we calculated it as:

$$\lambda_{\_alt} = \lambda_{\_surface} * \frac{P_{-\,surface}}{P_{-\,alt}} * \frac{\text{Sampling time}_{\_alt}}{\text{Sampling time}_{\_surface}} * \varepsilon \qquad (16)$$

where $\lambda_{\_surface}$ is the $\lambda$ quantified experimentally at the surface.

Since air pressure and sampling times were directly measured, the primary source of error for $M_{\_alt}$, and consequently $\lambda_{\_alt}$, arises from pump efficiency ($\varepsilon$), which may decrease over time and at lower pressures. Using the estimated $\lambda_{\_alt}$, the observed vertical isotope profiles were corrected based on Eq.14 from Section 2.2. The uncertainty estimation is discussed in Section 3.4.2."

L306 comment 1: what does "diffusion model correction process" mean? Parameterization? Correction of model structure?

We corrected 'Potential sources of error in the diffusion model correction process' to 'Potential sources of error in correcting vertical observations using the diffusion model' (line 365).

L306 comment 2: I don't think lambda_surface and lambda_alt are proper variables. Lambda is a property of a bag that should not depend on altitude. Its apparent dependence on altitude in this work is due to errors estimating masses. Therefore, unless I am missing something, the better variable to report here as a source of error is the estimate of air mass.

We completely agree with the reviewer. The variation in $\lambda$ at different altitudes is not due to an inherent altitude dependence but rather results from errors in estimating M. We have revised the manuscript to emphasize that $\lambda_{\_alt}$ is derived directly from $\lambda_{\_surface}$, pressure, sampling time, and pump efficiency ($\varepsilon$), as shown in Eq.16 (Section 3.4.1):

$$\lambda_{\_alt} = \lambda_{\_surface} * \frac{P_{-\,surface}}{P_{-\,alt}} * \frac{\text{Sampling time}_{\_alt}}{\text{Sampling time}_{\_surface}} * \varepsilon$$

Since air pressure at different altitudes is well-defined and sampling time is accurately recorded, these parameters do not contribute to uncertainty. Instead, the primary source of error stems from variations in pump efficiency ($\varepsilon$), which directly affects $M_{\_alt}$ and, consequently, $\lambda_{\_alt}$.

Therefore, we now explicitly identify the uncertainty in $\lambda_{\_alt}$ is fully propagated from the uncertainties in $\lambda_{\_surface}$ and $\varepsilon$, rather than being treated as an independent source of error. We have revised the manuscript accordingly to reflect this clarification (lines 344-363). The full details of the corresponding revision can be found in our response to the comment above.

L306 comment 3: mismatches between model and data are not sources of error, they are themselves the error.

We appreciate the reviewer's clarification. We have revised the manuscript to distinguish error sources from model-data mismatches. For example, in lines 365-366:

*"Potential errors in correcting vertical profiles using the diffusion model include estimates of $\lambda_{\_surface}$, $\alpha$, Sampling time$_{\_alt}$, and mismatches between model and experiments (Table 1)."*

Sec 3.4.2 the uncertainty section is difficult to follow and needs a revision for conciseness and clarity. The section includes too much information (e.g., how mean parameter estimates were obtained—i.e., nothing to do with uncertainty), is not well organized, and is also not always specific when it needs to be. An example of this last point is the ¼ estimate for pumping time. Is this really uncertain to that degree? It seems more likely (lacking actual experimental details), that pumping time is well known and the real variable is mass captured.

We appreciate the reviewer's feedback and have revised Section 3.4.2 for improved clarity and conciseness. We have removed unrelated details while retaining necessary information and reorganized it for better readability (lines 364-390):

*"3.4.2 The method of uncertainty estimation*

*Potential errors in correcting vertical profiles using the diffusion model include estimates of $\lambda_{surface}$, $\alpha$, pump efficiency ($\varepsilon$), and mismatches between model and experiments (Table 1). We detail each below:*

*1) $\lambda_{surface}$ uncertainty: laboratory experiments provided upper and lower bounds on $\lambda_{surface}$ (Subsection 4.1). These were used for error estimation.*

*2) $\alpha$ uncertainty: the $\lambda$ and $\lambda/\alpha$ were first estimated from several experiments, from which $\alpha$ was calculated. Their averaged values were used separately to parameterize the model. As highlighted in Subection 3.2.2, estimating $\lambda/\alpha$ (and subsequently calculating $\alpha$) requires results from cases where $q_0$ equals $q_e$, minor variations in $q_0$ and fluctuations in $q_e$ could introduce non-systematic discrepancies between the model and experimental results. Consequently, for analyzing the contribution of $\alpha$ to uncertainties, only $\alpha$ values derived from experiments where the model closely matched the majority of experimental results were considered. Selection criteria for these experiments included minimal deviation between $q_0$ and $q_e$, minimal deviation between experimental data and simulations, and stable $q_e$ values, ensuring the reliability of the chosen $\alpha$.*

*3) Pump efficiency ($\varepsilon$) uncertainty: The efficiency of the pump may decline over time or vary with atmospheric pressure, affecting the collected air mass M. To account for this, we applied a conservative uncertainty range of 0.75 to 1.25 relative to surface conditions, ensuring the full range of possible variations in $M_{alt}$ was considered.*

*4)Model-experiment mismatches: We compared model simulations with experimental data across 87 cases, calculating the average absolute discrepancy. These mismatches were included as an additional uncertainty component.*

*Total Uncertainty Calculation: The maximum discrepancy across all calibration results—using the full uncertainty range for $\lambda_{surface}$, $\alpha$, and pump efficiency ($\varepsilon$)—was determined. The model-experiment mismatch was then added as an independent error component. The final uncertainty estimates, reported in Subsections 4.3 to 4.5, account for all potential error."*

L376 this temperature (and pressure) dependence should be recognized in the theoretical development.

We have revised the text to explicitly acknowledge the temperature and pressure dependence (lines 433-440):

*"The specific parameter values obtained in this study pertain to the Teflon air bags used in the aforementioned tests, conducted at an ambient temperature of 16°C. These values depend on bag material, temperature, and pressure, which should be considered when applying the model under different conditions. We also noted batch-to-batch variations among air bags from the same manufacturer. We apply α measured under ground-level storage and measurement conditions, assuming negligible temperature and pressure effects during the short (10–20 min) drone-based sampling period. Future work is needed to quantify these dependencies."*

Fig 3 it would improve the accessibility if the caption told us which experiment these data come from

We have revised the figure caption to specify the corresponding experiments (line 441):

"Figure 3 Determination of 3 parameters of the diffusion model : $\lambda_{surface}$ (a) from Experiment No. 1, $\alpha\_^{18}O$ (b), and $\alpha\_^{2}H/$ (c) from Experiment No. 2."

Fig 4a misspelling of Environment. Also, Environment should be defined in the main legend, not in each panel

We have corrected the misspelling of "Environment" in Fig. 4a and moved its definition to the main legend:

[Figure]

Fig 4 I don't understand why the isotopic equilibration models cross over each other at long time. What equation exactly generates the solid lines model fits?

We appreciate the reviewer's question. We now provide an interpretation of this unexpected result. To achieve this, we slightly modified the final form of Eq. 8 to isolate

two terms: the first term drives $R_{(t)}$ towards $R_e$ at a constant rate, while the second term drives $R_{(t)}$ towards $R_e/\alpha$ at a rate dependent on $(q_e-q)$ (line 163).

[revised manuscript text omitted]

L398 should specify HD16O—or omit it, since the sentence is about 18O

We have revised HDO to HD$^{16}$O for accuracy.

Fig 5 delete "(a, b) (a-b)" at the beginning of the caption

We have removed it.

Fig 5 is difficult to follow. What is "real value"--it looks like initial isotopic composition inside the bag, but why does it not change with time and why is it measured at different times compared to the colored dots? What equation exactly generates the solid lines model fits?

We have revised "real value" to "Reference: initial values inside the air bag" for clarity. This value is also plotted at different times to serve as a reference for comparing changes in the air bag over time. We have also revised the legend of Fig. 5 accordingly.

As stated in our response to comments on Fig. 4, we have provided a relevant description in the manuscript (lines 198-211 and 301-309) regarding the equation used to generate the lines model fits. Additionally, we have explicitly clarified the meaning of the lines and markers in the caption of Fig. 5 (line 467):

For figures 4 and 5, plotting the humidity inside the bags over time would help a lot in illustrating the processes and ensuring the models are describing the processes correctly.

We appreciate the reviewer's suggestion. Figure 4a already presents humidity inside the bags over time, and we have additionally provided this information for Figures 5 and 6 in Supplementary Figure S2.

We have also clarified this in the manuscript (lines 447-450):

*"To validate the model, we used Experiment No.3 described in Subsection 3.2. The simulations from our diffusion model (lines in Figs. 4, 5, 6, and Fig. S2) are in close agreement with our experimental observations (markers in Figs. 4, 5, 6, and Fig. S2), showing consistency in humidity, $\delta^{18}O$, $\delta^2H$, and d-excess variations, with only minor deviations."*

For figures 5 and 6, I appreciate the idea to diagram the processes, but I found the diagrams unhelpful because they illustrate only the magnitudes of fluxes and ignore the crucial differences in humidity inside the bag.

To clarify the role of humidity differences inside and outside the air bag, we have revised the figure captions to explicitly highlight the impact of $q_0$ vs. $q_e$ on isotopic evolution (lines 467 and 490):

*"Figure 5    (a-b) Variations of $\delta^{18}O$ under different conditions: (a) when both the differences between internal ($\delta^{18}O_0$) and external ($\delta^{18}O_e$) $\delta^{18}O$ values as well as between internal ($q_0$) and external ($q_e$) humidity are not significant, $\delta^{18}O$ gradually increases toward equilibrium; (b) when $q_0$ is significantly lower than $q_e$, a stronger vapor influx causes enhanced kinetic fractionation, leading to a decrease in $\delta^{18}O$. (c-d) Corresponding schematics: (c) illustrates the mechanism for (a), where a weaker humidity gradient results in slower isotopic shifts, while (d) corresponds to (b), showing intensified fractionation with a larger gradient, with arrows indicating vapor flux direction and fractionation intensity. $\delta^{18}O$ (t) is the variation of $\delta^{18}O$ within the air bag over time. In (a) and (b), the colored lines represent diffusion model simulations based on Eq.8, using parameterization from Experiments No. 1 and 2. The square-marker-connected lines indicate the initial values inside the air bag, which remain constant over time and serve as a reference for comparison (legend: Reference)."*

*"Figure 6 (a-b) Evolution of d-excess in cases: (a) when the difference between the humidity inside ($q_0$) and outside ($q_e$) the air bag is not significant, d-excess increases gradually; (b) when $q_0$ is significantly lower than $q_e$, a stronger vapor influx enhances kinetic fractionation, causing a more rapid d-excess increase. (c-d) Corresponding schematics: (c) illustrates the mechanism for (a), where a smaller humidity gradient results in slower isotopic shifts, while (d) corresponds to (b), showing intensified fractionation with a larger gradient. $d_0$ indicates the initial d-excess value at t = 0, $d_e$ represents the d-excess in the environment. d(t) denotes the variation of d-excess within the air bag over time."*

L405 referring to Fig 4 as the first scenario and Fig 5 as the second and third scenarios is confusing, because they are not so labeled in the figures and difficult to keep straight in the text.

We appreciate the reviewer's feedback and have revised the text to improve clarity. Instead of referring to Figures 4, 5 and 6 as specific scenarios, we now explicitly describe the conditions presented in each figure to avoid confusion.

L431 again it would be helpful to be explicit about which equation is "the model"

We have provided a relevant description in the manuscript (lines 198-211 and 301-309) regarding what 'the model' refers to. The full details of the corresponding revision can be found in our response to the comment on Fig. 4.

L441 ok but (1) there are other ways to flag for unrealistic results, so focusing on this one seems odd; and (2) leaving in those six data points would presumably not have

much effect on the results, so this detail seems distracting.

We have removed the following sentences from the manuscript and retained data points with d-excess values less than 1‰:

"In this dataset, acquired from the drone observations and subsequently corrected using the diffusion modeling, data points with d-excess values less than 1‰ were omitted, as these values are unrealistic and likely result from overcorrection of the δ-values. This resulted in the exclusion of 6 out of 1039 samples."

This revision only affects a very small number of data points (6 out of 1039), and including these data does not alter the overall results and conclusions.

Minor global comment: the word "value" is redundant almost everywhere.

We have reviewed the manuscript and removed redundant instances of "value" for conciseness while retaining necessary ones for clarity.

L456 "model corrections" can never affect d18O in the bag. I think this is saying that applying corrections for vapor pressure differential and fractionation by the bag changes the estimate of the atmospheric d18O. L454-459 why say all this twice? The correction process is the same for 2H and 18O, and the d response follows. This description makes it sound more complicated than that.

We appreciate the reviewer's clarification and have revised the text to eliminate redundancy and improve clarity (lines 526-531):

*"The strong kinetic fractionation driven by the diffusion of air into the air bag results in a decrease in the water vapor $\delta^{18}O$ within the bag. After applying model corrections, the corrected $\delta^{18}O$ values inside the bag increased slightly compared to pre-correction levels. As described in Subsection 4.2, vapor flux with higher d-excess entering the bag increases the d-excess inside. As a compensation, the diffusion model applies corrections, resulting in a reduced d-excess value after correction (Fig. 7c and 10)."*

Fig 10 it is not clear what "Picarro" means here. Is this the measurement of ambient vapor at the surface at the time of sampling aloft? Or is this bag samples vs. satellite-inferred estimates?

We have revised the figure caption (line 576) and related descriptions (lines 577-579):

*(line 576) "(a, b) Raw and corrected (with uncertainties) altitude-averaged air bag measurements from 3856 m to 4000 m, compared with in-situ surface-level measurements at 3856 m taken by the Picarro (legend: Picarro)."*

*(lines 577-579) The left panel of Figure 10 (Fig. a, c, and e) shows the comparison of raw and corrected water vapor $\delta^2H$ measurements at different altitudes with in-situ surface-level measurements on the Picarro or IASI satellite data at corresponding altitudes.*

L551 the methods in this manuscript have nothing to do with laser spectrometry; the samples could as well be measured by other methods.

We appreciate the reviewer's clarification and have revised the text to remove unnecessary references to laser spectrometry (lines 629-632):

*"Our drone-based sampling system, combined with the diffusion model, effectively addresses the limitations of traditional high-altitude water vapor measurement methods. It meets the need for lightweight equipment while providing a more economical, efficient, and flexible alternative to conventional approaches involving large aircraft, airships, and balloons."*

---

## Author Response (AR2)

**Response to the Editor**

Dear Dr. Janssen,

We would like to thank you very much for your decision and for the constructive comments provided in your editorial note. We carefully revised the manuscript according to your requests and the minor suggestions. Below we provide a detailed point-by-point response.

Our responses are in blue, with specific changes to the text highlighted in *blue italics*.

All line numbers in this document correspond to the line numbers in the updated version of the manuscript.

With best regards,
Di WANG
On behalf of all authors

Dear Authors,

Thank you for submitting your article to AMT and for making the changes requested by the referees of a previous round. Based on these, and two more recent independent reviews, I decide publishing your article subject to minor revisions.

However, in order to assure good quality I urge you to address all of the referee's questions, particularly the request to put the results into perspective and to discuss the practicability of the correction/method (especially in the case when the correction goes into the wrong direction). Please also ensure that you use coherent notation and provide complete definitions in your revision and, in this context, introduce delta and d-excess values by giving the definition of these quantities so that a link can be made between the measurements/modelling (based on R) and the observational data given in the figures.

We appreciate the editor's decision to publishing our article subject to minor revisions.

We have carefully addressed all reviewer and editor comments to further improve clarity, consistency, and practical context.

Ensuring consistent notation throughout the manuscript, particularly for λ, α, and other model parameters.

We now provide the full definitions of δ and d-excess (lines 230–235):

*"The δ and d-excess values used in this study follow standard definitions:*

$$\delta^{18}O = \left( \frac{R^{18}O_{sample}}{R^{18}O_{standard}} - 1 \right) * 1000 \qquad (15)$$

$$\delta^{2}H = \left( \frac{R^{2}H_{sample}}{R^{2}H_{standard}} - 1 \right) * 1000 \qquad (16)$$

$$d - excess = \delta^2 H - 8 * \delta^{18}O \qquad\qquad (17)$$

Here, $R^{18}O_{standard} = 2.0052*10^{-3}$ and $R^2H_{standard} = 1.5576 * 10^{-4}$, *corresponding to the VSMOW international reference values."*

Please prepare the revised manuscript by further taking into account the following three points apart from the minor corrections suggested in the list below.

1) According to Figure 7, it appears that diffusion corrections on d18O tend to be more

relevant than corections on D, ie the correction at high altitudes has the size of the assigned error bar for 18O whereas the error bar for D is much larger than the correction itself. Therefore, d18O needs to be displayed in Fig 10 as well. This would make it easier to judge the quality and usefulness of the proposed diffusion correction.

We appreciate the editor's suggestion. Figure 10 presents comparisons with IASI satellite data, which currently provide retrievals only for $\delta^2H$, not for $\delta^{18}O$. To our knowledge, no existing satellite product offers $\delta^{18}O$ retrievals for atmospheric water vapor. Therefore, $\delta^{18}O$ cannot be included in this comparison. At the same time, we believe this limitation highlights the value of our drone-based sampling approach, which enables vertical profiling of $\delta^{18}O$, a capability that is currently inaccessible via satellite remote sensing.

To clarify this limitation, we have added the following statement in the methods section (lines 428-430):

*"In this study, as no satellite retrievals of $\delta^{18}O$ are currently available, we compared our observed vapor $\delta^2H$ profiles up to the upper troposphere with satellite observations."*

2) The fact that lambda depends on bag size seems to contradict the model assumptions that the diffusion occurs across the surface, which should depend on the bag material only. What does the fact that lambda differs significantly between 0.5L bags and 4L bags (see values in Table 1) imply ? Does this question your conclusions ?

We thank the editor helping us clarify a potentially confusing point. Our theoretical formulation (Eq. 6) shows that the water vapor exchange coefficient $\lambda$ is proportional

to the exchange surface area $A$ of the air bag: $\lambda = \frac{k*A}{M}$. Therefore, $\lambda$ is expected to vary

with bag size, as larger bags typically have greater surface areas. The observed differences in $\lambda$ values between the 0.5 L and 4 L bags are fully consistent with this theoretical relationship. We clarified this in the methods section (lines 278-281):

*"As $\lambda$ is related to the exchange area (surface area of the air bag), measurements were conducted using 0.5L and 4L air bags, with repetitions on both identical and different air bags of the same dimensions (refer to the experiment times in Table 1 and results in Fig.3a)."*

3) The fact that bags are not perfect and have finite permeability raises the question on how diffusion into the bags is taken care off before the bags are actually "filled". It seems that such effect can be neglected (eventually by continuous pumping on the

empty bag?), but the paper does not say so and the reconstitution equation starts at the time t=0 of the filling at measurement altitude. Can you be more specific on the sampling conditions and the diffusion modeling concerning this part ?

*We have now clarified the sampling process in the revised manuscript (lines 333-347):*

*"We designed and built a collection module for fixed-height sampling, incorporating diaphragm vacuum pumps, a rudder mounted on the drone, and a control module linked to a remote operating system. Before flight, each air bag was evacuated on the ground using the diaphragm pumps. When inactive, these self-sealing pumps effectively isolated the interior of the air bags from the external environment, minimizing diffusion prior to sampling. When the drone reaches a specified altitude, we remotely activate the designated air pump to inflate a specific air bag. Once sampling is complete, the pump is deactivated, and the drone ascends to the next target altitude, where the corresponding air pump inflates another air bag. This process was repeated until all predetermined samples were collected. After each sampling, the pump remained sealed, ensuring no unintended air ingress during descent., preventing unintended air ingress. Additionally, due to the flexible nature of the air bags, internal and external pressures remained balanced. As air pressure increases during the drone's descent after collection. To further prevent the loss of collected air samples, a one-way valve was installed to block backflow. Additionally, the one-way valve helps prevent large droplets from entering the air bag during the collection process."*

*Given these measures, we are confident that diffusion into the bags prior to filling was negligible.*

Minor suggestions

l. 138 : the definition of kinetic fractionation factors is typically k_heavy/k_light. Please follow the convention and explain that you are approximating k_heavy ~ k.

*We thank the editor for this helpful suggestion. In the revised manuscript, we have clarified this point as follows (lines 148-150):*

*"Since k represents the total flux including both light and heavy molecules, it is approximated as the conductance of the heavy molecules ($k_{heavy}$), so that Eq. (3) is consistent with the conventional definition of fractionation factors ( $\alpha = \frac{k_{heavy}}{k_{light}}$). "*

l. 204 : For clarity, please complete the phrase "... and the ambient values (qe(t), Re18O(t) and ReD(t)) at all times during and after the flight "

*The revised sentence now reads (lines 211-214):*

*"If we know the initial values within the air bag ($q_0$, $R_{18}O_0$, $R_2H_0$), the ambient values during storage ($q_e$, $R_{18}O_e$, $R_2H_e$; approximated using ground-level conditions), and the storage duration after sampling ($T_{\_storage}$), we are able to simulate the variations in humidity and isotopic ratios inside the air bag according to Eqs. 5 and 8."*

l. 231 : The policy of our journal is to use SI units throughout. If you want to refer to

your instrument reading/setting in non-SI units, you may give the value in SI (or related units such as hPa/mbar) and repeat the non-SI value (psi) in parantheses.

We have revised the sentence to comply with the SI unit policy by converting the pressure values to kilopascals (kPa) and retaining the non-SI unit (psi) in parentheses. The revised text reads (lines 249-251):

*"In the measurement procedure, we first activated the dry air cylinder and adjusted the pressure reducing valve to approximately 13.8 kPa (2 psi), within the Picarro water isotope analyzer's recommended range of 13.8–27.6 kPa (2–4 psi) for carrier gas."*

l. 241, "we achieved greater acccuracy" This is not evident. Please check your phrase. While precision is generally improved through averaging repeat measurements, accurray only improves if the accuracy is limited by statistical noise and not by stystematic bias.

Thank you for this remark. Our intention was not to claim an improvement of accuracy, but rather to emphasize that replicate samples were used to ensure the consistency of the measurements. We have therefore revised the sentence to (lines 259-261):

*"By repeatedly measuring isotopic composition for replicate samples—air samples collected simultaneously under the same conditions—we ensured the consistency of the water vapor measurements."*

l. 393. Satellite isotope data IASI -> Satellite isotope data from IASI

Revised to "Satellite isotope data from IASI".

l. 365 The method of uncertainty estimation -> Uncertainty estimation

We thank for your suggestion. To help readers clearly distinguish between the methodological description in Section 3.4.2 ("The method of uncertainty estimation") and the results presented in Section 4.4 ("Uncertainty estimates"), we would prefer to retain the original sub-section title.

Table 1 Average of all difference between ... -> Average of all differences ...

We have corrected "difference" to "differences".

l. 535 differentsources -> different sources

We have corrected this to "different sources".

l. 536 We analyzed ... -> We represent ... .

We have revised the phrase from "We analyzed …" to "We represent …".

l. 536ff. Please clarify your approach further by recalling the meaning of the pdf-peak positions and -widths.

We are grateful to the reviewer for prompting this clarification. We have added an explanation of the PDF structure at the beginning of the paragraph, which improves the readability and interpretability of the subsequent analysis (lines 559-563):

*"We represent the contributions of uncertainty sources to vertical vapor δ18O and d-excess measurements at different altitudes using probability density function plots (Fig. 8). Each PDF peak indicates the most probable uncertainty value, while its width reflects sensitivity to that parameter. Narrow PDFs suggest stable, well-constrained uncertainty sources, whereas broader PDFs indicate greater variability and a more diffuse impact on the correction outcome."*

In addition, we have updated the caption of Figure 8 to include this clarification (line 558):

*"Figure 8: Contributions of different sources of uncertainty for $\delta^{18}O$ (a–c for 4000 m, 6000 m, and 9000 m, respectively) and d-excess (d–f for 4000 m, 6000 m, and 9000 m, respectively). The green triangle indicates the average mismatch between the model and experimental results. The peak position of each probability density function (PDF) represents the most likely uncertainty value for a given source, while the width of the distribution reflects variability."*

l. 566 The combined uncertainty from all sources, including λ_surface, α, pump efficiency (ε), and model-experiment mismatches, results in a total uncertainty ... ->
The combined uncertainties from all sources, including λ_surface, α, pump efficiency (ε), and model-experiment mismatches, result in a total uncertainty ... ->

We have corrected the grammar and revised "results" to "result".

l. 571 Additionally, λ_surface and α and also contributes considerably ... Additionally, λ_surface and α and also contribute considerably

We have removed redundant "and".

Figures 8 and 9. Please take care of y-axis legends that have been cut off.

We have corrected the y-axis labels in Figures 8 and 9.

**Response to Reviewer #2**

General comments:
Wang et al. used a drone and inflatable teflon bags to sample air from different altitudes (3856 -11000 masl) for subsequent, ground-based water vapor isotope measurements. Raw data were corrected for assumed, environmentally induced isotope-fractionating diffusion through the sampling bags' walls. Parameters needed for the diffusion model had been obtained from temperature-controlled laboratory experiments simulating some of the environmental conditions encountered in an 11 km air column. The attempt to find a widely applicable, low-cost solution for aerial sampling for water vapor isotope analysis is appreciated, but the results still need to be put into the right perspective. Great efforts were made in this study to estimate diffusion parameters and apply the correction model but parts of the respective description could perhaps go to the attachment or supplemental material in order to streamline the manuscript.

We sincerely thank the reviewer for the positive evaluation of our method and for the constructive suggestions. We will address the specific comments point by point carefully. To ensure that all necessary content remains clearly presented, we did not make major structural changes to the manuscript.

It stands out from the comparison of corrected data with independent observations (in situ for near-ground sampling, IASI satellite data for higher altitudes) that the applied correction model often fails to significantly eliminate the observed offsets between raw bag data and the respective reference observations, especially in the case of high-altitude sampling. In many cases, correction-induced shifts of data were even in the wrong direction, i.e. away from reference observations, which makes the entire correction approach questionable or at least insufficient. This holds even for near-ground samples, where the environmental conditions should be known best and exposition of samples to potentially adverse conditions should be shortest between collection and analysis. Further, a quantitative assessment of the offset between corrected data and reference observations is critically needed and must be compared to the data uncertainty acceptable for atmospheric water vapor isotope studies.

We thank the reviewer for raising this important point.

Near-surface case (Fig. 10a–b): Because only a few bag samples were collected at surface, our original plot also included samples up to 4500 m. We have now updated the figure to show only the 3856 m bag samples, allowing a direct comparison with the in-situ Picarro observations at the same altitude. The in-situ Picarro values are period-averaged, whereas the bag samples are instantaneous, which may explain their minor differences. Quantitatively, the mean absolute errors (MAE) after correction are 0.51 ‰ for $\delta^{18}O$, 2.99 ‰ for $\delta^2H$, and 1.37 ‰ for d-excess, all within the range acceptable for atmospheric water vapor isotope studies.

Higher altitudes (Fig. 10c–f): For $\delta^2H$, comparisons were made with IASI satellite retrievals, which provide values only for three broad vertical layers (1–3 km, 4–7 km, 8–12 km). These data represent vertical averages determined by averaging kernels and therefore allow only qualitative comparison with observations. This limitation

highlights the importance of our drone-based high-resolution profiles. As expected, the corrections behave consistently with physical principles: the adjustments for $\delta^{18}O$ are small because diffusion effects are relatively minor, whereas for $\delta^2H$ and especially d-excess the corrections are larger, reflecting the stronger kinetic fractionation of hydrogen isotopologues.

Importantly, the detailed uncertainty assessment provided in this study effectively compensates for the limited availability of independent reference data for direct quantitative comparison. Our method provides a detailed assessment of all potential sources of error and quantifies the uncertainty range of the corrected data- including $\lambda\_surface$, $\alpha$, pump efficiency ($\varepsilon$), and model-experiment mismatches—is ~1 ‰ for $\delta^{18}O$ and ~8 ‰ for d-excess across 98% of the data. These ranges are slightly larger than the precision of direct Picarro in-situ measurements (~0.5 ‰ for $\delta^{18}O$ and ~4 ‰ for d-excess), but remain acceptable when compared to the substantial vertical variations we observed (~20‰ for $\delta^{18}O$ and ~100 ‰ for d-excess). We have now made this quantitative comparison explicit in the revised text (lines 592-595):

*"The combined uncertainty from all sources, including $\lambda\_surface$, $\alpha$, pump efficiency ($\varepsilon$), and model-experiment mismatches, results in a total uncertainty of approximately 1‰ for $\delta^{18}O$ and 8‰ for d-excess across 98% of the data. These ranges is acceptable when compared to the substantial vertical variations we observed (~20‰ for $\delta^{18}O$ and ~100 ‰ for d-excess)."*

The other detailed discussion and the corresponding manuscript revisions are provided in our reply to Fig. 10 and L578.

In general, I appreciate the changes that have been made in response to previous revisions and I agree with the authors that a low-cost solution for high-altitude, high-resolution air sampling would benefit the community. However, I cannot second their conclusion that the presented approach is a big step forward in this direction (expressed by, e.g., "notable agreement", "practical solution" etc.). Currently, the main obstacle seems to be the choice of a not-diffusion-tight sampling bag, the too simplistic assumptions made regarding the diffusion model or other significant processes beyond diffusion that were not considered in the applied correction (potentially varying temperature and pressure conditions).

We thank the reviewer for this balanced comment. In the revised manuscript, we clarified the limitations of the current approach and its applicability.

In the conclusion we now state (lines 670-673):

*"This study was conducted under stable temperature and equal internal and external pressures during storage. We acknowledge that a formulation based on partial pressure of water vapor would be more general and could improve model applicability under varying temperature and pressure conditions. Future work could extend the model to account for these factors."*

In addition, we explicitly noted this in the main text:

*(e.g., lines 458-460) "These values depend on bag material, temperature, and pressure, which should be considered when applying the model under different conditions."*

We also carefully toned down expressions such as "notable agreement" and "practical solution" to more cautious wording.

Specific Comments:

L 71 and L81 The cited study by Jiménez-Rodríguez is only available as preprint in HESSD. It was under review in 2019 but never accepted. No revision was provided by the authors afterwards, thus no accepted peer-reviewed version exists. Therefore, this pre-print must not be cited in your manuscript.

Instead, you could add Havranek et al. (2020), Magh et al. (2022), Herbstritt et al. (2023) in L71, and Gralher et al. (2021) in L 81.

We thank the reviewer for catching this. The preprint has been removed. We now cite Havranek et al. (2020), Magh et al. (2022), Herbstritt et al. (2023) in L71 and Gralher et al. (2021) in L81, as suggested.

L 95-97 The bags used by Herbstritt et al. (2023) were proven diffusion-tight. Issues with these impermeable bags were based on adsorption not on diffusion. Please rephrase accordingly.

We thank the reviewer for pointing this out. We have revised the sentence to clarify that Herbstritt et al. (2023) used diffusion-tight bags and that the remaining challenges were mainly related to adsorption rather than diffusion (lines 95-97):

*"To mitigate these issues, specific diffusion-tight materials are highly recommended for water vapor isotope measurements, although adsorption effects may still occur with such materials (Herbstritt et al., 2023)."*

L 107-108 Varying humidity and differences in isotopic composition is important, but what about varying (partial) pressure and varying temperature? This will eventually also affect material properties and diffusion coefficients of the Teflon bags in up to 11000masl. An easy pre-test would have been to fill some replicates of the bags, put them on the drone and take them to high altitudes just for exposure to the different ambient conditions, then bring them down again for analysis and compare the results with measurements from other replicate bags, which had been stored at the ground, as well as with the in situ measurements. Was something similar done?

We appreciate this insightful suggestion. The actual residence time of our samples at high altitude was very short. We did not directly perform exposure-only high-altitude experiments, as conducting controlled tests at different altitudes, temperatures, and pressures is technically very challenging. Moreover, even if air bags were transported to ~11 km altitude and then brought back to the surface, as suggested by the reviewer, the bags would still not only be exposed to a stable high-altitude environment. For these reasons, we consider our approach both needed for done-based observation and practical: as described in Section 3.4 (Eqs. 15–16), we estimated $\lambda$ ($k*A/M$) for different altitudes ($\lambda_{alt}$) through the variable of collected air mass (M), which was calculated from pressure, sampling time, and pump efficiency ($\varepsilon$). A comprehensive uncertainty assessment was carried out to account for the potential variability of these factors.

Regarding pressure, since the bags are flexible, no net pressure gradient exists across the membrane during sampling and storage. For temperature, we simplified the treatment by assuming isothermal laboratory conditions. In reality, samples remained at high altitude only briefly (10–20 minutes). Moreover, the lower ambient temperature at high altitudes and the reduced difference between ambient air and the stored samples compared to ground-level conditions are expected to further limit the magnitude of diffusion effects. Therefore, we have applied conservative error ranges in our uncertainty analysis, which has already overestimated the true impact of these factors.

We clarified that while our current treatment does not explicitly resolve the temperature dependence of bag permeability, this represents an important avenue for future work to improve the general applicability of the model:

In the Introduction (after line 119–126):

*"However, because of the short residence time at high altitudes, the reduced diffusivity at low temperatures, and the flexibility of the bags maintaining equal internal and external pressure, our current approach does not explicitly resolve potential temperature- or pressure-dependent changes in bag permeability. Under our experimental conditions, these effects were instead addressed indirectly through conservative uncertainty estimates. Future applications should take into account the specific experimental conditions, and future developments may extend the model to explicitly incorporate the dependence of bag permeability on partial pressure and temperature, thereby improving its applicability under a wider range of atmospheric conditions."*

L 121ff. Fickian diffusion as a physical law has been described before. Please cite the general equation(s) accordingly. Same for 'isotopic fractionation'.

We thank the reviewer for pointing this out. We have now clarified in the text that Eq. (1) is derived from Fick's first law of diffusion (Fick, 1855) (lines 136-140):

*"The flux of water into the bag, F (in kg/m2/s), is expressed as:*

$$F=k*(qe-q(t)) \tag{1}$$

*where q(t) represents the variation of humidity inside the air bag over time (in kg/kg), qe denotes the environmental humidity (in kg/kg), k is water vapor conductance. This first-order formulation is derived from Fick's diffusion law(Fick, 1855)."*

Figure 10: Comparison of corrected data with independent observations shows that the applied correction model often fails to significantly eliminate the observed offsets, especially in the case of high-altitude sampling (10c and 10e). It stands out that in many cases, even the direction of the applied correction is wrong, i.e. raw data plot between corrected data and references, rendering the applied model or its parameters questionable. This holds even for near-ground sampling (10a and 10b), where no significant isotopologue-specific vapor pressure gradients must be assumed to affect the respective samples' integrity during the short time between collection and analysis. I assume that a better agreement of independent observations with raw bag data rather than corrected data would also become evident when presenting data in cross plots including a 1:1 line (in addition to the time series), which I strongly suggest.

We thank the reviewer for this valuable suggestion. We have addressed the deviations between corrected and reference data in our general reply. For the near-surface case, because only a few bag samples were collected at 3856 m, our original plot also included samples up to 4500 m. We have now updated the figure to show only the 3856 m bag samples, allowing a direct comparison with the in-situ Picarro observations at the same altitude.

Following the reviewer's advice, we have also added cross plots comparing raw and corrected air bag data with in-situ observations averaged over the sampling period, including a 1:1 reference line, in the revised manuscript. For $\delta^2H$, the corrected data fall essentially along the 1:1 line, while for d-excess the points are more scattered. Nevertheless, our method explicitly quantifies the uncertainty range of the corrected values. As shown in this figure (Fig.A3), and as expected for near-surface sampling, the differences between the bag samples and the ambient values are very small, and the impact of diffusion is minimal. Consequently, the applied corrections are also small, although they bring the data closer to the 1:1 line. Overall, the raw bag values, the corrected bag values, and the in-situ Picarro measurements remain in very close agreement.

L 578 What measure was used for the assessment of 'notable agreement'? Please provide quantitative assessments of the offsets between 'raw measurements' vs. references and 'corrected data' vs. references and compare them to the uncertainty acceptable for atmospheric water vapor isotope studies.

We thank the reviewer for this helpful suggestion. We have now quantified the offsets between corrected data and in-situ Picarro time-averaged observations at 3856 m using the mean absolute error (MAE). After correction, the MAEs are 0.51 ‰ for $\delta^{18}O$, 2.99 ‰ for $\delta^2H$, and 1.37 ‰ for d-excess. These values are well within the combined uncertainty of our correction method (~1 ‰ for $\delta^{18}O$ and ~8 ‰ for d-excess across 98% of the data) and are comparable to the precision typically considered acceptable for atmospheric water vapor isotope studies. We have added this quantitative statement in the revised manuscript (lines 607-611).

*"There is agreement between the raw and corrected $\delta^2H$ measurements for altitudes 3856–4000 m and the in-situ $\delta^2H$ observed directly by the Picarro at ground level (3856 m), with a mean absolute error of 2.99 ‰ (Fig. 10a, Fig. A3). The remaining minor differences can be attributed to the fact that the Picarro in-situ observations are period-averaged, whereas the bag samples represent instantaneous values."*

*"After correction, d-excess decrease and are similar to in-situ surface-level measurements on the Picarro at 3856 m, with a mean absolute error of 1.37 ‰."*

The mentioned dependencies of the specific parameters on temperature and pressure as well as the mentioned batch-to-batch variations (L 435-440) could also be reasons for the observed deviation between the 'measured data' and the 'Picarro' data in Fig. 10 and should be briefly discussed here.

This has been addressed in our response above and corresponding changes have been made in the manuscript

Technical corrections:

L 2 Typo: should be 'case studies' instead of 'cases studies'.

"Cases studies" has been corrected to "case studies."

L 23 Please delete the comma.

Thank you. We have removed the comma.

L43 Typo: 'in situ' instead of 'in suit'.

Corrected. "In suit" has been replaced with "in situ".

L 86 and throughout the manuscript: please check the order of the references (oldest to newest).

We have carefully revised the reference order throughout the manuscript, in accordance with the journal's guidelines.

L 137 Please add '(\alpha)' after '…fractionation coefficient' and check for identical notation (sometimes ',\alpha,') throughout the manuscript.

We have added "α" after "fractionation coefficient" as requested, and ensured that the notation for α is consistently applied throughout the manuscript.

L 153 Please add '(\lambda)' after '…exchange coefficient'.

"λ" has been added after "exchange coefficient"

L 184 \alpha is the fractionation coefficient, not the mass flow. Please move the '\alpha' accordingly.

We thank the reviewer for pointing this out. We have revised the sentence accordingly to avoid the misleading association between α and mass flow (lines 194-196):

"This equation demonstrates that the slope of ln ($R_e$-$R(t)$) against time is the water vapor isotopic exchange coefficient $\frac{\lambda}{\alpha}$. Knowing λ, we can deduce the isotopic fractionation coefficient α for each isotope."

Figure 2 The air bag in the picture seems to be metalized – is this a Teflon bag??

We have updated Figure 2 with a clearer photo showing the actual Teflon air bag used in this study.

L 233 Gas flow rate is '40sccm at 760 Torr' according to the L2130-i Analyzer Datasheet.

We thank the reviewer for the clarification. In the revised manuscript, we have updated the description to be consistent with the L2130-i Analyzer datasheet (lines 251-254):

"The instrument's built-in flow regulation maintains a gas flow rate of 40 sccm at 760 Torr, corresponding to ~ 30–50 mL/min in our measurement conditions, ensuring

*stable sample delivery."*

L244 the correct name is 'L2130-i'.

 We have corrected "Picarro 2130i" to "Picarro L2130-i".

Figure 10 is still hard to understand. 'Measured data' are also measured by the Picarro, right? To avoid confusion, I would suggest something like 'Raw air bag data' – 'Corrected air bag data' – 'in situ measurements' in the figure legend.

 We agree and appreciate the helpful suggestion. We have revised the legend of Figure 10.

Literature:
Havranek et al. (2020) https://doi.org/10.1002/rcm.8783
Magh et al. (2022) https://doi.org/10.5194/hess-26-3573-2022

---

## Author Response (AR3)

**Response to the Editor**

Dear Dr. Janssen,

We sincerely thank you for your decision and for the attention to detail. We have carefully revised the manuscript according to your requests and implemented all suggested technical and formatting improvements to ensure consistency with AMT standards. Below we provide a detailed point-by-point response.

Our responses are in blue, with specific changes to the text highlighted in *blue italics*.

All line numbers in this document correspond to the line numbers in the updated version of the manuscript.

With best regards,
Di WANG
On behalf of all authors

Dear Mr Wang, dear authors,

I would like to thank you again for submitting to AMT and I also thank you for responding to the referee's and my requests. I appreciate the great detail that you have dedicated to each of your responses that helped to clarify most of the concerns. Nevertheless, there remains one issue which has not been addressed satisfactorily. This requires that you go once again through a revision of your manuscript.

The comparison of your bag measurements with independent methods is a substantial part of the paper. However, as already indicated in the last referee report

"It stands out from the comparison of corrected data with independent observations (in situ for near-ground sampling, IASI satellite data for higher altitudes) that the applied correction model often fails to significantly eliminate the observed offsets between raw bag data and the respective reference observations, especially in the case of high-altitude sampling. In many cases, correction-induced shifts of data were even in the wrong direction, i.e. away from reference observations, which makes the entire correction approach questionable or at least insufficient. This holds even for near-ground samples, where the environmental conditions should be known best and exposition of samples to potentially adverse conditions should be shortest between collection and analysis."

it remains unclear whether the correction has a significant impact on comparisons with other platforms, whether in situ or satellite. What readers want to know is whether it

affects the comparison between the bag data and the other instruments. Does it improve or worsen the comparison? For example, the data in Fig. A3 are too scattered to actually 'see' the effect of the correction. Clear evidence is needed to show whether the diffusion correction improves the comparison with other observations, such as a statistical analysis based on the correlation coefficient or residuals showing how the agreement between the bag data and the validation platform changes with the correction. The results of this analysis should be discussed critically in light of the referee's remark cited above. The same applies to the comparison with satellite data. Please check all your statements regarding this comparison. The current discussion is inappropriate, particularly the statement in the abstract that 'the corrected observations match the Picarro in situ observations and IASI satellite retrievals'. According to panel c of figure 10, where both the uncorrected and corrected data are substantially higher than the observations, the statement is both incorrect and misleading, as the graph suggests that the correction has no significant effect on the agreement between the bag and the IASI data.

We sincerely thank you for this constructive follow-up. We fully understand that the key remaining issue concerns whether the diffusion correction significantly improves the agreement between the air-bag measurements and the other dataset (Picarro and IASI).

In this revised version, we have (1) performed a quantitative statistical analysis (correlation coefficients and mean absolute errors), (2) clarified that the comparison with IASI is qualitative only, and (3) note the difference in the upper troposphere, and critically discussed the sources of discrepancy and the limitations of satellite representativeness.

For the comparison between the ground-level bag measurements and the in-situ Picarro observations, the correction was indeed small for both $\delta^2H$ and d-excess, which is expected because the air inside and outside the bags had nearly identical humidity and isotopic composition. As described in Section 2.1 (Eq. 8), the diffusion effect depends on the humidity and isotopic difference between the inside and outside of the bags.

We have accordingly revised the text to provide clearer quantitative comparisons and explanations (lines 614-626):

*"At near-ground level, both the raw and corrected $\delta^2H$ and d-excess values from the bag measurements show close agreement with the in-situ Picarro observations (Fig. 10a and b), indicating minimal diffusion effects under nearly identical humidity and isotopic conditions between the inside and outside of the bags. Most data points are distributed along the 1:1 line (Fig. A3), while d-excess shows slightly larger scatter. After applying the correction, the agreement improved: the correlation coefficient between the bag and Picarro $\delta^2H$ values increased from 0.90 to 0.99, and for d-excess from 0.41 to 0.72. The mean absolute error (MAE) between the corrected bag and Picarro measurements is 3.0‰ for $\delta^2H$ and 1.4‰ for d-excess. These discrepancies are within the uncertainty range derived from laboratory diffusion experiments and model– experiment mismatches, which were comprehensively incorporated in the error*

*estimation (Sections 3.2 and 3.4): 0.5 ‰ for $\delta^{18}O$, 4.1 ‰ for $\delta^2H$, and 2.9 ‰ for d-excess. The differences also could be due to the fact that the Picarro in-situ observations are period-averaged, whereas the bag samples represent instantaneous values."*

For higher altitudes, we can only rely on satellite data IASI, but as described in the Methods section, the IASI dataset is not strictly comparable with the local high-resolution observations. The MUSICA retrievals from the IASI satellite instrument provides water vapor isotope data at three altitude levels: 1-3 km in the lower troposphere, 4-7 km in the mid-troposphere, and 8-12 km in the upper troposphere. However, these measurements represent a vertical average over layers determined by the averaging kernels. While applying the averaging kernels to smooth our observed profiles would allow a more quantitative comparison, the multi-level data with averaging kernels for 2020 are not publicly available and were still inaccessible at the time of our delayed response submission. Therefore, we averaged our observations over the corresponding altitude ranges; as a result, the comparison remains mainly qualitative. The effect of the averaging kernels on altitude could contributes to the differences between IASI and our observations in the upper troposphere, which also highlights the importance of our high-resolution vertical measurements.

In addition, even if the IASI or our observational profiles were adjusted according to the averaging kernels, three other factors would still further limit their comparability, which we now explicitly discuss below:

(1) Temporal mismatch: IASI provides only two overpasses per day (local overpass around 09:30 am and 09:30 pm), whereas our observations were made during daytime. Diurnal variability can introduce significant differences (Herman et al., 2014; Lacour et al., 2018). Moreover, IASI data are not available for every observation date, even at locations close to our sampling site.

(2) Spatial mismatch: The IASI footprint (~12 km at nadir) represents a coarse spatial average, whereas our measurements are local and instantaneous. Therefore, the satellite retrievals cannot fully represent the fine-scale variability at our sampling site.

(3) Most important—Cloud-sampling limitation: IASI cannot observe through or below thick clouds, such as anvil clouds. Consequently, it mainly samples the environment outside convective detrainment regions. Since convective detrainment isotopically enriches vapor (Kuang et al., 2003; Moyer et al., 1996; Smith et al., 2006; Vries et al., 2021; Webster and Heymsfield, 2003), the air masses observed by IASI—being farther from detrainment—are expected to be more depleted. This sampling limitation could therefore contribute to the lower $\delta^2H$ values in the upper troposphere (around and above cloud top) retrieved by IASI compared with our in-situ observations.

The diffusion of air bags should result in systematic bias. But IASI data sometimes agree well with our measurements, sometimes show higher values, and sometimes lower values (Fig. 10). These fluctuations arise from representativeness differences inherent to the satellite retrievals, not from diffusion-related biases. Therefore, such variations cannot be corrected using the diffusion model, which accounts for isotopic fractionation during air-bag storage but not for spatiotemporal (both horizontal and

vertical) discrepancies between measurement platforms, nor for the observational limitations of satellites regarding cloud processes.

We have now included the uncertainty ranges of the IASI retrievals in Fig. 10, which just account for retrieval—fit noise and atmospheric temperature a priori constraint. The uncertainty range is already large. In the 4000–7000 m range, it almost entirely covers our observation range, demonstrating good consistency within errors. However, in the upper troposphere (7000–12000 m), the IASI data still appear lower than our observations. This discrepancy likely arises primarily from cloud-sampling limitations, as well as from temporal and spatial representativeness mismatches, rather than from diffusion-related biases. We explicitly acknowledge this difference in the revised Discussion and thoroughly analyze all potential causes (lines 627-676):

[revised manuscript text omitted]

In Section 4.2, we demonstrate that our diffusion model successfully reproduces the experimental results under a wide range of environmental conditions, confirming its robustness and its ability to quantify the influence of diffusion on water vapor isotope measurements.

As emphasized in the Conclusion, an important contribution of this study is the methodological guidance it provides:

"We recommend prioritizing the use of glass containers and air bags with the lowest permeability for collecting water vapor using portable devices. Additionally, it is essential to conduct the permeability experiments described in this article before any experimental undertaking. This involves storing water vapor with known isotopic composition in the portable collection device for an extended period and then re-measuring these values to assess or determine the device's permeability parameters." This recommendation represents a key outcome of our work, offering a practical standard for future field measurements.

Regarding the vertical profiles, although no completely independent dataset is available for direct validation, our model—already verified by laboratory experiments—was used to propagate all potential error sources and to calculate a conservative and comprehensive uncertainty range.

In the revised manuscript, we no longer emphasize detailed matching with the IASI data, but rather highlight the three main contributions of this work:

(1) methodological recommendations for future isotopic vapor sampling,
(2) development of a robust diffusion model validated under various experimental conditions
(3) quantitative uncertainty assessment for the vertical profiles.

Apart from this point, I would like to ask you to consider the following minor comments to revise your manuscript.

General technical comments:

There are several occurrences of underscores in the text and formulae, such as λ_surface or M_alt. This is somewhat non-standard and reads like computer code. Could you please remove the underscore when you use the subscript in combination with a mathematical symbol or variable such as (lambda, alpha, etc.)

Thank you for the suggestion. We have removed the underscores and replaced them with proper subscripts throughout the text and equations.

Please check that the quality of graphs is sufficient and that exes labels are uncut (see Fig. 10 for example).

Thank you for your reminder. We have checked all figures to ensure sufficient quality and that all axis labels are fully visible.

Detailed comments:

l 42 "The corrected observations match the Picarro in-situ observations and IASI satellite retrievals." -> This statement is not only wrong (see above), but it is also misleading.

We have deleted this sentence, we no longer emphasize detailed matching with the IASI data, but rather highlight the three main contributions of this work:
(1) methodological recommendations for future isotopic vapor sampling,
(2) development of a robust diffusion model validated under various experimental conditions
(3) quantitative uncertainty assessment for the vertical profiles.

l 119 "However, because of the short residence time at high altitudes, the reduced diffusivity at low temperatures, and the flexibility of the bags maintaining equal internal and external pressure, our current approach does not explicitly resolve potential temperature- or pressure-dependent changes in bag permeability. Under our experimental conditions, these effects were instead addressed indirectly through conservative uncertainty estimates. Future applications should take into account the specific experimental conditions, and future developments may extend the model to explicitly incorporate the dependence of bag permeability on partial pressure and temperature, thereby improving its applicability under a wider range of atmospheric

conditions."

"However, because" sounds strange.

You could instead write something along the following lines "Although we minimised certain diffusion-related biases, such as pressure differences and long storage times, temperature and pressure effects on bag permeability were only addressed indirectly via conservative uncertainty estimates. Future applications should consider the specific experimental conditions and extend the model to explicitly incorporate the dependence of bag permeability on partial pressure and temperature. This would improve the model's applicability to a wider range of atmospheric conditions."

*Thank you for the suggestion. We have revised the paragraph accordingly as follows (lines 118-127):*

*"Although we minimized certain diffusion-related biases, such as pressure in different altitudes and long storage times, temperature and the pressure difference between the inside and outside of the bags on bag permeability were only addressed indirectly via conservative uncertainty estimates. This treatment is justified under our experimental conditions, which are characterized by the short residence time at high altitudes, the reduced diffusivity at low temperatures, and the flexibility of the bags maintaining equal internal and external pressure. Future applications should consider the specific experimental conditions, and future developments may extend the model to explicitly incorporate the dependence of bag permeability on partial pressure and temperature, thereby improving its applicability under a wider range of atmospheric conditions."*

l 148 Here, I need to apologize, because I was obviously wrong in one of my earlier remarks on the definition of alpha. According to the "IUPAC gold book" (https://goldbook.iupac.org/terms/view/K03405), the standard definition of alpha is k_light/k_heavy. Also, your approximation is k = k_light. Please change your phrases accordingly.

*We sincerely thank the editor for this clarification. In our previous revision, we mistakenly followed an earlier suggestion without carefully rechecking our own definition, which led to an incorrect expression of $\alpha$. We have now corrected this and confirmed that the current formulation is fully consistent with the IUPAC definition ($\alpha$ = k_light / k_heavy). To facilitate checking, the relevant section is shown below, only the revised sentences are marked in light blue italics (lines 149-151):*

*"The flux of water into the bag, F (in $kg/m^2/s$), is expressed as:*

$$F = k * (q_e - q(t) \tag{1}$$

*where q(t) represents the variation of humidity inside the air bag over time (in kg/kg), $q_e$ denotes the environmental humidity (in kg/kg), k is water vapor conductance. This first-order formulation is derived from Fick's diffusion law(Fick, 1855).*

*Similarly, the flux of isotopologue, $F_i$, either $H_2^{18}O$ or HDO, moving into the bag can be described as:*

$$F_i = k_i * (R_e * q_e - R(t) * q(t)) \tag{2}$$

*In this equation, $k_i$ represents the conductance specific to each isotopologue (in*

kg/m$^2$/s), R$_e$ denotes the isotopic ratio in the environment, and R(t) is the variation of isotopic ratio within the air bag with time. Notably, the fractionation coefficient here can be denoted as:

$$\alpha = \frac{k}{k_i} \tag{3}$$

*Since k represents the total conductance dominated by the lighter molecules (H$_2^{16}$O), and k$_i$ is defined as the conductance of the heavier isotopologues (either H$_2^{18}$O or HDO), Eq. (3) is therefore consistent with the conventional definition of fractionation factors ( α=k$_{light}$/k$_{heavy}$)."*

l 278 I suggest to write "As lambda is related to the surface area of the sampling bag, " ...

   We have modified as suggested.

l 342 delete ", preventing unintended air ingress.", as this has already been said in the sentence before

   Thank you. We have deleted as suggested.

l 388 "The method of uncertainty estimation" -> "Method of uncertainty estimation"
   We have replaced the sentence as recommended.

Fig 3 "α_δ 18O" should be "α18O" and "α_δ2H" should be "α2H"
   Thank you for pointing out the missing updates. We have corrected them accordingly.

l 556 "determined the ranges for four errors to evaluate uncertainty" -> "investigated four error sources to evaluate the uncertainty"
   We have replaced the sentence as recommended.

l 559 Consider to replace "We represent the contributions of uncertainty sources to vertical vapor δ18O and d-excess measurements at different altitudes using probability density function plots (Fig. 8)." -> "We represent the contribution of each source of uncertainty to vertical vapor δ18O and d-excess measurements at different altitudes using probability density function plots (Fig. 8)."
   We have replaced the sentence as recommended.

l 599 Common confidence intervals are 68 %, 95.5 % or 99.7 %, which corresponds to 1, 2 or 3 sigma, respectively, if a standard normal probability distribution function (Gaussian) is assumed. Maybe you could give your result once for 95% and once for the 98% confidence interval.

   Thank you. We have added the corresponding 95% confidence intervals (~0.9‰ for δ$^{18}$O and ~6.9‰ for d-excess) as suggested.

l 594 "these ranges is" -> "this range is"
   We have revised the grammar.

l ~604, figure caption: delete "(legend: in situ measuremeents)"

Thank you. We have deleted as suggested.

Fig. 8, If the mismatch is represented as a pdf, it is a delta-function with infinite amplitude so that the integral is 1. Please correct in the picture by drawing a vertical line which goes to the upper axis of the graph. Also the unit of the ordinate axis must be 1/‰.

Thank you for your suggestion. We have modified Fig. 8 accordingly by adding a vertical line to represent the mismatch as a delta function and corrected the ordinate unit to 1/‰.

l 609 change "2.99" to "3" or to "3.0", because I don't think that your result is significant to 2 digits after the decimal point.

We have adjusted the decimal place of the error value.

l 627 "After correction, the a mean absolute error is 1.37 ‰. ." -> "After correction, the mean absolute error is 1.4 ‰."

Thank you. We have adjusted the decimal place of the error value and removed the extra "a".

ll 630 I strongly recommend replacing "As previously noted, raw d-excess are higher than corrected data due to kinetic fractionation. After correction, d-excess decreases." by "As previously noted, corrected d-excess values are lower than the raw data."

Thank you. We have replaced the sentence as recommended.

l 654 check sentence: ", and offers" instead of ", offers" ?

Thank you. We have modified as suggested.

Fig A3. Panel b) shows the wrong data. While δ18 should be presented, the data belong to measurements of δ2H. Please display the correct data.

Thank you for your comment. The axis label was mistakenly written as $\delta^{18}O$. We have now corrected the label to $\delta^2H$.

---

## Author Response (AR4)

Dear Dr. Janssen,

Thank you very much for your careful review. We confirm that all requested technical corrections have been made exactly as specified, including:

- Line 150: placing the equation inline and removing the space after the parenthesis, as requested: ($\alpha = k_{light}/k_{heavy}$).
- Line 520: correcting "Fig.5b and d" to "Fig. 5b and d".
- Line 568: correcting "four errors sources" to "four error sources".
- Line 655: replacing the sentence:
  *"However, it is noteworthy that the IASI data closely match the observed $\delta^2H$ for all other periods in the 4000–7000 m range, except for June 2020, where they are lower too."*

We have carefully checked the full manuscript to ensure consistency and accuracy following these adjustments.

With kind regards,
Di Wang (on behalf of all co-authors)